# Automated model optimisation using the Cylc workflow engine (Cyclops v1.0)

Richard M. Gorman[1], Hilary J. Oliver [2]

[1] National Institute of Water and Atmospheric Research, PO Box 11-115, Hamilton, New Zealand
[2] National Institute of Water and Atmospheric Research, Private Bag 14901, Wellington, New Zealand

*Correspondence to*: Richard M. Gorman (Richard.Gorman@niwa.co.nz)

**Abstract.** Most geophysical models include many parameters that are not fully determined by theory, and can be 'tuned' to
improve the model's agreement with available data. We might attempt to automate this tuning process in an objective way by
employing an optimisation algorithm to find the set of parameters that minimises a cost function derived from comparing
model outputs with measurements. A number of algorithms are available for solving optimisation problems, in various
programming languages, but interfacing such software to a complex geophysical model simulation, presents certain challenges.
To tackle this problem, we have developed an optimisation suite ("Cyclops") based on the Cylc workflow engine that
implements a wide selection of optimisation algorithms from the NLopt Python toolbox (Johnson, 2014). The Cyclops
optimisation suite can be used to calibrate any modelling system that has itself been implemented as a (separate) Cylc model
suite, provided it includes computation and output of the desired scalar cost function. A growing number of institutions are
using Cylc to orchestrate complex distributed suites of interdependent cycling tasks within their operational forecast systems,
and in such cases application of the optimisation suite is particularly straightforward.

As a test case, we applied the Cyclops to calibrate a global implementation of the WAVEWATCH III (v4.18) third generation
spectral wave model, forced by ERA-Interim input fields. This was calibrated over a one-year period (1997), before applying
the calibrated model to a full (1979-2016) wave hindcast. The chosen error metric was the spatial average of the root-mean-
square error of hindcast significant wave height compared with collocated altimeter records. We describe the results of a
calibration in which up to 19 parameters were optimised.

## 1 Introduction

Geophysical models generally include some empirical parameterisations that are not fully determined by physical theory, and
which need calibration. The calibration process has often been somewhat subjective and poorly documented (Voosen, 2016),
but in a more objective approach has the aim of minimising some measure of error quantified from comparisons with
measurement (Hourdin et al., 2017). We can turn this into an optimisation problem: namely to find the minimum of an objective
function $f(\vec{x})$ where $\vec{x}$ represents the set of adjustable parameters, and $f$ is a single error metric (e.g. the sum of RMS differences
between measured and predicted values of a set of output variables) resulting from a model simulation with that parameter set.
The most efficient optimisation algorithms require the derivative $\vec{\nabla} f(\vec{x})$ to be available alongside $f(\vec{x})$. This, however, is rarely
the case for a geophysical modelling system, so will restrict our attention to the field of Differential Free Optimisation (DFO),
in which the objective function $f$ can be calculated, but its gradient is not available.
Various methods exist, many of which are summarised in the review of Rios and Sahinidis (2012). Some are good at exploring
parameter space to improve the likelihood of finding global rather than merely local minima. Others are preferred for quickly
moving to the absolute minimum once in its neighbourhood. The algorithms are encoded in various languages (e.g. Fortran,
C, Python, Matlab), and usually require the user to supply a subroutine to compute $f(\vec{x})$, that can be called as required by the
optimisation programme.

This is satisfactory for many problems where the objective function is readily expressed as an algorithm, but is somewhat less straightforward to interface an existing geophysical model, as well as all the methods needed to process and compare measurement data with an optimisation code, in this way. Nevertheless, examples of this approach can be found in hydrological and climate modelling applications. For example, Seong et al. (2015) developed a calibration tool (using R software) to apply the Shuffled Complex Evolution optimization algorithm to calibrate the Hydrologic Simulation Program-Fortran (HSPF) model. In climate modelling, Severijns and Hazeleger (2005) used the downhill simplex method to optimize the parameter values of the subgrid parameterizations of an atmospheric general circulation model. More recently, Tett et al. (2013) applied a Gauss–Newton line search optimization algorithm to climate simulations with the Hadley Centre Atmosphere Model version 3 (HadAM3) forced with observed sea surface temperature and sea ice, optimising an objective function derived from reflected shortwave radiation and outgoing longwave radiation comparisons. The Tett et al. (2013) method was subsequently applied to optimize the sea ice component of the global coupled HadCM3 climate model (Roach et al., 2017;Tett et al., 2017).

Such custom applications of one particular optimisation algorithm to a specific model, however, can require significant effort to switch to alternative optimisation algorithms, or to be applied to new models. Modern coupled climate models, or operational forecast systems for weather and related processes, encompass a diverse set of software tools, often running on multiple platforms. Ideally, we would like to be able to optimise performance of the modelling *system* (not just a single model code) without major reconfiguration of software between the calibration and operational/production versions of the system.

The Cylc workflow engine is now applied in several operational centres to manage the scheduling of tasks within such systems. So it seems natural to consider the possibility of developing a framework within Cylc for the optimisation of the modelling systems under its control.

## 2 Methods

In very general terms, a derivative-free optimisation algorithm will explore parameter space, selecting values of the parameter vector $\vec{x}$ in some sequence. As each $\vec{x}$ is selected, it calls the (user-supplied) subroutine to evaluate the objective function $f(\vec{x})$. In our case, this would amount to running a complete model simulation with the corresponding parameter settings, comparing outputs to measurements, from which a defined error metric is computed to provide the return value of $f$. This can involve a lengthy simulation, needing a run time $T_{model}$ perhaps of order hours or days to reproduce months or years of measurements.

A self-contained optimisation program, with an explicitly-coded function-evaluation subroutine, will run much faster, with a run time per iteration $T_{iter}$ typically being some small fraction of a second, and will run in many orders of magnitude less time than a typical geophysical model even if a number of iterations $N$ of order 1000 are required. This might be the case for "deliberately difficult" test problems: we might expect that a well-tested geophysical model will come with reasonable defaults that in many new implementations will produce a result within a relatively simple "basin of attraction" so that O(10) iterations may suffice to get very close.

If the optimisation procedure calls for a full model run to evaluate the objective function, and $N$ iterations are required for convergence, the total run time would be

$$T \approx T_0 + N(T_{model} + T_{iter}) \qquad (1)$$

including an overhead $T_o$ for initial and final tasks.

As $T_{model}$ is orders of magnitude larger than $T_o$ and $T_{iter}$, the geophysical modelling system totally dominates run time, and we can comfortably afford not to be concerned with reducing the efficiency of the optimisation routine, even by a few orders of magnitude.

So let's consider a simple measure we might introduce to allow us to recover from an interruption part way through a long optimisation process. Normally, the optimisation code will retain in memory the values of each $\vec{x}$ and its objective function $f(\vec{x})$ that has already been evaluated, to use in selecting further points to be evaluated. If we write these values to file each time

the function evaluation is called, we can build up a lookup table to use in case we need to restart the process. In that case, we could have the function evaluation subroutine first search the lookup table for a match to $\vec{x}$ (within some acceptable tolerance), in which case it could return the tabulated error value. Only in the case where a tabulated value was not found would the full model simulation be required to compute the return value of $f$.

Now rather than actually perform that computation, the function evaluation subroutine could simply write the $\vec{x}$ values (for the $n^{\text{th}}$ iteration, say) to file, and exit. We could then run the model in its usual way, outside the optimisation code, using those $\vec{x}$ values as parameters, and add that result to our lookup table before restarting the whole process from scratch. This time, assuming the optimisation algorithm is deterministic, with no random process influencing the sequence of $\vec{x}$ values, the first $n$ points would be exactly the same sequence that was selected previously, and could be quickly handled by table lookup, and

the algorithm would either find that a convergence criterion had been satisfied, or select a new point $n+1$ to be passed to the model for simulation.

In effect, we are simply employing the optimisation algorithm in a generic tool that, given the results of all previous iterations, either signals that convergence has been reached, or generates the next parameter set to be evaluated by the model, i.e.

$$\vec{x}_{n+1} = Opt(\{\vec{x}_m, f_m\}_{m=1,\ldots,n}) \tag{2}$$

In this scheme, assuming that we start with an empty lookup table, the first pass has one iteration of the optimisation code, the second has two, etc. So, allowing an additional overhead $\hat{T}$ for the full process, the total run time to reach the termination condition(s) after $N$ iterations should be

$$T' = \hat{T} + \sum_{n=1}^{N}(T_o + nT_{iter} + T_{model}) \tag{3}$$

$$= \hat{T} + N(T_o + T_{model}) + \frac{N(N+1)}{2}T_{iter} \tag{4}$$

As $T_{model}$ is orders of magnitude larger than the other times, the ratio of the two run times is

$$\frac{T'}{T} \approx 1 + \frac{N+1}{2}\frac{T_{iter}}{T_{model}} \tag{5}$$

Given the expected relative magnitudes of the model and optimisation iteration times, and $N$ of order 10s or 100s, the increase

in run time through this approach is actually negligible.

On the other hand, this scheme has several benefits. Apart from being simple to code, the optimisation algorithm, including the user-defined function evaluation subroutine, can be completely generic, and applied unmodified to different modelling systems. The only requirements on the modelling system are that, at the start of each simulation, it reads in the parameter values requested by the optimisation code and adapt them to its standard input formats, then at the end of the simulation,

computes and writes to file a single error metric value. The optimisation code and the model system could then remain separate, both controlled by some form of scripting scheme, for example. This means that having invested considerable time and resources in developing a complex modelling scheme, no major reconfiguration needs to be made to prepare it for optimisation in this manner, or subsequently to re-implement the optimised modelling system in operational or production mode.

## 2.1 Cylc

Cylc (http://cylc.github.io/cylc/) is an Open Source workflow engine that can manage ongoing distributed workflows of cycling (repeating) tasks. It was originally developed at NIWA to automate environmental forecasting systems, and has since been adopted by many other institutions – notably the UK Met Office and its international partners in climate, weather, and related sciences. Cylc can manage large production systems of great complexity, but it is also easy to use for individuals and groups with less demanding automation requirements. Cylc workflows (or *suites*) are defined with an efficient graph syntax

that expresses dependence between tasks, and an efficient inheritance hierarchy for optimal sharing of all task runtime properties (exactly what each task should execute, and where and how to submit task jobs to run).

Cylc tasks are related by *trigger expressions* that combine to form a *dependency graph*. This trigger,

   A:*status* => B

says that task B depends on task A achieving the status *status* ("=>" represents an arrow). The default trigger status is *succeeded* (job execution completed successfully) and can be written simply as A => B; others include *submitted, submit-failed, started, finished, failed*, and custom task output messages. Tasks can depend on the wall clock and on external events, as well as on other tasks, and a task job can be submitted to run once all its dependencies are met. Cylc automatically wraps user-defined task content (environment, scripting, etc.) in code to trap errors and report job status back to the suite server program via

authenticated HTTPS messages. Tasks can even trigger off tasks in other suites, so for coupled systems you can choose between a larger suite that controls all tasks, and multiple smaller suites that interact with each other.

In cycling systems tasks repeat on sequences that may represent forecast cycles, or separate chunks of a model simulation that is too long for a single run, or iterations in an optimization scheme, or different datasets to be processed as they are generated, and so on. Cycling is specified with ISO 8601 date-time recurrence expressions (e.g. for environmental forecasting), or with

integer recurrence expressions (e.g. for iterative processes). Both date-time and integer cycling are used in the application described in this paper. Dependence across cycles (consider a forecast model that is initialised with outputs from a previous cycle) creates ongoing, potentially never-ending, workflows. Uniquely, Cylc can manage these without imposing a global cycle loop: one cycle does not have to complete before the next can start. Instead, tasks from many cycles can run concurrently to the full extent allowed by individual task dependencies and external constraints such as compute resource and data

availability. So, for example, on restarting after extended downtime, a suite that processes real-time data can clear its backlog and catch up again very quickly, by automatically interleaving cycles.

## 2.2 Implementation

We have developed a Cylc suite ("Cyclops", https://zenodo.org/badge/latestdoi/1836229) to perform optimisation of a modelling system that has itself been set up as a separate Cylc suite. In the example we describe below, the model suite controls

a multi-year wave model hindcast, including the preprocessing of necessary model inputs (principally wind fields) and verification data (satellite altimeter data), running the wave model code, postprocessing of model outputs, and generation of error statistics from comparisons of predicted and observed significant wave height fields.

Typically, date-time cycling is used to run a model at successive forecast cycles, or to break a long simulation into a succession of shorter blocks. The optimisation suite, on the other hand, uses integer cycling, with each cycle corresponding to a single

evaluation of the objective function.

There are several tasks controlled by the optimisation suite. One of these is responsible for running an optimisation algorithm to identify either an optimal parameter vector from previous model runs, or the next parameter vector to be evaluated. This main optimisation task within the suite is implemented with Python code calling the NLopt Python toolbox (Johnson, 2014).

NLopt includes a selection of optimisation algorithms: both "local" solvers, which aim to find the nearest local minimum to

the starting point as efficiently as possible, and "global" solvers, which are designed to explore the full parameter space, giving high confidence in finding the optimal solution out of a possible multitude of local minima. NLopt includes algorithms capable of using derivative information where available, which is not the case in our application, and Cyclops is restricted to the derivative-free algorithms listed in Table 1.

We have assumed that the sequence of parameter vectors tested by an optimisation algorithm is deterministic. Several of the

algorithms available in NLopt have some inherently stochastic component. It is, however, possible to make these algorithms "repeatably stochastic" by enforcing a fixed seed for the random number generator.

In NLopt, any combination of the following termination conditions can be set:

1. maximum number of iterations by each call of the optimisation algorithm

2. absolute change in the parameter values less than a prescribed minimum

3. relative change in the parameter values less than a prescribed minimum

4. absolute change in the function value less than a prescribed minimum

5. relative change in the function value less than a prescribed minimum

6. function value less than a prescribed minimum

In the second and third of these convergence criteria, the "change in parameter values" means the magnitude of the vector difference, i.e. $\sqrt{\sum_{n=1}^{Npar}(\Delta x_n)^2}$.

We have implemented Python code that uses NLopt calls to seek a minimum of an objective function $f$ that represents a non-negative model error metric. As described above, the user-defined function evaluation has been implemented as a generic Python function $f(\vec{x})$ that simply searches a lookup table (stored in a file). If $\vec{x}$ is found in the table it returns the corresponding $f$ value, otherwise it saves the vector $\vec{x}$ to a file and returns an invalid[1] $f$ value. Any of the termination conditions listed above can be set by the user: the last of these can use a prescribed minimum $f$ value as a convergence condition, while an invalid $f$ value signals that the optimisation algorithm has stopped because a new parameter vector $x$ needs to be evaluated externally by a model simulation. In this case a file is written containing parameter names and values in a format that can be parsed by the modelling system to generate the needed input files for a simulation. At present a generic namelist format is used as output from Cyclops for this purpose.

A "parameter definition" file is used to specify parameter names and their initial values, as used within the model. If a parameter is allowed to be adjusted by the optimisation suite, an allowable range is also set. This choice will generally require some experience with the particular model. Within the optimisation suite, these adjustable parameters will be scaled linearly to normalised parameters $\vec{x}$ that lie between 0 and 1. Fixed parameters can be include for convenience, so that their names and values will be written to the namelist file but these are ignored by the optimisation suite.

The major tasks carried out by Cyclops on each cycle are:

0. (first cycle only): **Init**: write initial normalised parameters $\vec{x}_0$ to file, …

1. **Optimise**: run the optimisation code, starting from $\vec{x}_0$ and evaluating every $\vec{x}$ in the sequence, until either a stopping criterion is met (in which case the task sends a "stop" message), or a parameter set $\vec{x}$ is reached that is not in the lookup table so needs evaluating (signalled by a "next" message)

2. **Namelist**: Convert $\vec{x}$ to non-normalised parameters in a namelist file

3. **Model**: Create a new copy of the model suite, copy the namelist file to it, and run it in non-daemon mode (i.e. so the task will not complete until the model suite shuts down). A new copy of the suite is made so that files created in one cycle do not overwrite those created on other cycles.

4. **Table**: Read the resulting error value from the model suite, and update the lookup table

Within one cycle, the dependencies of the optimisation suite are simply:

      **Optimise**:next => **Namelist** => **Model** => **Table**

---

[1]At present $f < 0$ is treated as an "invalid" return value, which is appropriate for positive-definite error metrics, but the Python code could be modified to return $f$ = None for more general cases.

to make these tasks run sequentially when no stop condition is met. We set a dependency on a previous cycle:

**Table**[-P1] => **Optimise**

(the notation –P1 denotes a negative displacement of one cycle period), to ensure that the lookup table is up to date with all previous results before starting the next optimisation cycle, and to prevent Cylc from running successive cycles concurrently.

The stopping condition is handled by

**Optimise**:stop => **Namelist_Final** => **Stop**

where the **Namelist_Final** task produces the final version of the namelist file, and the **Stop** task does a final wrap-up of the completed optimisation before the suite shuts down. For the purposes of good housekeeping, we can also add a **Model_delete** task to delete each copy of the model suite once all its outputs have been used. Also, tasks which will not be needed (e.g.

"Namelist" if "Optimise" gives a "stop" message) can be removed, along with any dependencies on those tasks, by so-called "suicide triggers". Figure 1 illustrates the workflow of the optimisation suite described above in graphical form.

The optimisation suite's Model task for each cycle is a proxy for a copy of the full model suite being run for the corresponding parameter set. The model suite is run in non-daemon (non-detaching) mode, so that the Model task does not finish until the suite that it represents runs to completion. Information passed between the suites consists of two simple files: a "namelist" file

containing parameter names and values written by the optimisation suite for the model suite, and an "error" file containing the single value of the error metric returned by the model suite.

The model suite needs to include a task to process the namelist file into the particular modelling system's standard input formats. Because the formats are highly model-specific, this task needs to be tailored for the particular model suite. For example, in our wave hindcast application described below, this task consists of a shell script which simply includes the

namelist file verbatim as part of an ASCII control file, which also has various timing parameters provided from environment variables. Namelists can include named groups of parameters, which may be helpful in this process in cases where these groups need to be treated differently (e.g. affecting different model input files for multiple coupled models and pre- and post-processing tasks within the model suite). However, if the namelist format proved inadequate to supply the needed information, this format could be changed within the optimisation suite to something more suitable. It should be stressed, though, that no

change should be needed to the main model codes: they can run as standard release versions under a separate task within the model suite.

## 2.3 Concurrent simulations

For some DFO algorithms, at least some parts of the sequence of vectors tested is predetermined, and independent of the function values found at those points. For example, BOBYQA (which we chose to use in the test application described below)

sets up a quadratic approximation by sampling the initial point, plus a pair of points on either side of it in each dimension. With $N$ parameters, the first $2N+1$ iterations are spent evaluating these $2N+1$ fixed points, regardless of the function values obtained there. In such situations, the function values for each of these points could be evaluated simultaneously.

This can be done within Cylc by allowing tasks from multiple cycles to run simultaneously. In practice, this means that multiple copies of the model suite are running simultaneously, to the extent allowed by resource allocation on the host machine(s). This

makes it imperative that a new copy of the model suite is made for each cycle.

If concurrent model simulations are allowed, this means that at any time there are a certain set of parameter vectors for which the function values are still being determined (we can call this the "active" set). We can add another parameter vector to that set if it will be selected by the optimisation algorithm regardless of the function values at the active parameter vectors.

We would clearly like to determine that without needing specific knowledge of how the particular optimisation algorithm

works. Instead we use a simple empirical method. To this end, we supplement the lookup table (of vectors already computed, with the resulting $f$ values) with a second table (the "active file") listing the active vectors. We have the function evaluation subroutine search for $\vec{x}$ first among the "completed" vectors, then among the "active" vectors. If it finds $\vec{x}$ among the active

vectors (for which $f$ is not yet known), it assigns $f$ a random positive value (in this application we don't re-initialise the random number generator with a fixed seed). Otherwise it writes $\vec{x}$ to file and returns an "invalid" $f$ value to force the optimisation algorithm to stop as usual.

The Python code controlling the optimisation algorithm has also been modified. Now when the active file is empty it will act as before, but if there are active vectors it will run a small set of repeated applications of the optimisation algorithm (Eqn 2), each of which will use a different set of randomised $f$ values for the active vectors. . That is, in the Optimise task for cycle $n + 1$ we evaluate

$$\vec{x}_{n+1}^{(q)} = Opt(\left\{\vec{x}_m, f_m^{(q)}\right\}_{m=1,\ldots,n}) \tag{6}$$

for a set of iterations $q = 1, \ldots, Q$, with

$$f_m^{(q)} = \begin{cases} f_m & \text{completed } m \\ \text{random} & \text{active } m \end{cases} \tag{7}$$

If these all result in the same choice of $\vec{x}_{n+1}$ to be evaluated, a "next" message is sent to trigger further tasks for this cycle as before, since this choice is independent of the results for the active parameter vectors. If not, we do not have a definite $\vec{x}_{n+1}$ to evaluate, and we must wait until at least one of the presently active simulations has finished before trying again, a "wait" message is sent. But clearly this does not mean that the optimisation is complete.

These repeated randomised applications of the optimisation algorithm are run sequentially within one cycle of the Optimise task, simply to determine if there is a unique parameter set with which further tasks for that cycle can be started, concurrently with Model tasks already running for other cycles. They do not themselves need to run in parallel.

We also need to consider how the Cylc suite dependency structure can accommodate concurrent simulations. This can be handled in two ways. In the first method we let the Optimise task fail when it determines a "wait" condition, and utilise Cylc's facility to retry failed tasks at specified intervals.

We also replace the dependency

**Table**[-P1] => **Optimise**

with the combination

**Optimise**[-P1] => **Optimise**

**Table**[-P*M*] => **Optimise**

where $M$ is a specified maximum number of concurrent simulations. This means that each cycle can first attempt to start a new model simulation as soon as the previous cycle's simulation has *started* and the $M$th previous simulation has *completed*. The "Optimise" task will keep retrying at intervals until it is able to give either a "stop" or "next" signal. This method has a simple workflow structure, illustrated in Figure 2, that does not change as $M$ increases.

A schematic illustration of how this might work is shown in Figure 3. Here we consider an application in which the optimisation algorithm uses predetermined values for the first five parameter vectors, after which each new parameter vector selected depends on all previous results (BOBYQA has this behaviour for a two-parameter optimisation). We also assume we have set $M \geq 5$. Hence in cycle 2, the Optimise task's randomised test shows that the same parameter vector will be chosen regardless of the outcome of the cycle 1 Model task, so that further cycle 2 tasks can start immediately. Similarly, the cycle 3 Model task does not need to wait for the active cycle 1 and 2 Model tasks to complete, and so forth up to cycle 5. But the cycle 6 Optimise task will detect that its choice of a parameter vector will depend on the results of the active Model tasks, so it will fail and retry. Under our assumptions it will not succeed until no other Model tasks are active, and this will remain the case for all subsequent cycles.

The second method, described in Appendix A and used in the tests described here, uses more complex dependencies and additional Optimise tasks, instead of a single retrying Optimise task. It is somewhat more efficient in that there is no need to wait on a (short) retry interval before determining if a new cycle can start, but the workflow is more complicated and its

complexity increases with *M*. Both methods achieve the same result, however: they both allow up to *M* model suites to run concurrently, rather than iterating through them in sequence.

It should be stressed that the optimisation code itself is simply run as a serial process in each case: it is simply required to produce the single set of parameters, if any, for the next model run given the known results of the completed simulations. As it checks that this parameter set is independent of the results of the presently active model runs without needing to know the actual results, no parallel processing is required within the optimisation code.

## 3 Application: a global wave hindcast based on ERA-Interim inputs

Here we describe a global wave simulation, using the WAVEWATCH III model (WW3), forced by inputs from the ERA-Interim Reanalysis, covering the period from January 1979 to December 2016. Such multi-year wave model simulations are a valuable means of obtaining wave climate information at spatial and temporal scales that are not generally available from direct measurements. It is rare for a particular location of interest to have a suitably long nearby *in situ* wave record, e.g. from a wave-recording buoy, to provide statistically reliable measures of climate variability on inter-annual time scales. And while satellite altimetry has provided near-global records of significant wave height that have been available for more than two decades, these have limited use for many climate applications for several reasons, including a return cycle that is too long to resolve typical weather cycles, limitations in providing nearshore measurements, and lack of directional information. Model simulations can in many cases overcome these limitations, but available measurements still play an essential role in calibrating and verifying the simulations.

In our case, one of the principal motivations for carrying out this hindcast is to investigate the role of wave-ice interactions in the interannual variability of Antarctic sea ice extent, which plays an important role in the global climate system. The ERA-Interim Reanalysis is a suitable basis for this work, providing a consistent long-term record, with careful control on any extraneous factors (e.g. changing data sources, or modelling methods) that might introduce artificial trends or biases into the records. While the ERA-Interim Reanalysis includes a coupled wave model, direct use of the wave outputs does not fully meet our requirements, which include the need for the wave hindcast to be independent of near-ice satellite wave, which were assimilated into the ERA-Interim Reanalysis. Hence we chose to carry out our own wave simulation, forced with ERA-Interim wind fields, but with no assimilation of satellite wave measurements.

### 3.1 Comparison of model outputs with altimeter data

Rather than being assimilated in the hindcast, satellite altimetry measurements of significant wave height were used as an independent source of model calibration. These were obtained from the IFREMER database of multi-mission quality-controlled and buoy-calibrated swath records (Queffeulou, 2004).

Swath records of significant wave height were first collocated to the hourly model outputs on the $1° \times 1°$ model grid. For each calendar month simulated, collocations were then accumulated in $3° \times 3°$ blocks of 9 neighbouring cells to produce error statistics, including model mean, altimeter mean, bias and root-mean-square error (RMSE), and correlation coefficient *R*. Spatial averages of these error statistics were taken over the full model domain between 65°S and 65°N (excluding polar regions with insufficient coverage).

The final error statistic used in the objective function was the spatially-averaged RMSE, normalised by the spatially-averaged altimeter mean, temporally averaged over the simulation period, excluding spinup.

### 3.2 WW3 parameters

For this simulation we used version 4.18 of the WAVEWATCH III (WW3) third generation wave model (Tolman, 2014). The model represents the sea state by the two-dimensional ocean wave spectrum $F(\vec{k}, \vec{x}, t)$, which gives the energy density of the wave field as a function of wavenumber $\vec{k}$, at each position $\vec{x}$ in the model grid and time $t$ of the simulation.

The spectrum evolves subject to a radiative transfer equation

$$\frac{\partial N}{\partial t} + \vec{\nabla}_x \cdot (\dot{\vec{x}}N) + \frac{\partial}{\partial k}(\dot{k}N) + \frac{\partial}{\partial \theta}(\dot{\theta}N) = \frac{S}{\sigma} \tag{8}$$

for the wave action $N(k, \theta, \vec{x}, t) = F(\vec{k}, \vec{x}, t)/\sigma(k)$, where $\sigma$ is the frequency associated with waves of wavenumber magnitude $k$ through the linear dispersion relation, and $\theta$ is the propagation direction. The dots represent time derivatives. The terms on the left hand side represent spatial advection, and the shifts in wavenumber magnitude and direction due to refraction by currents and varying water depth. The source term $S$ on the right hand side represents all other processes that transfer energy

to and from wave spectral components, including contributions from wind forcing, energy dissipation and weakly-nonlinear four wave interactions.

Adjustable parameters within WW3 that can influence a deep water global simulation such as the one described here are principally concentrated in the wind input and dissipation source terms. It is generally necessary to treat these two terms together as a self-consistent 'package' of input and dissipation treatments designed to work together. In this study we undertook

two separate calibration exercises, based on two 'packages' of input/dissipation source terms, firstly that of Tolman and Chalikov (1996) (activated in WW3 by the ST2 switch), and secondly the Ardhuin et al (2010) formulation (using the ST4 switch).

In Appendix B we describe some of the details of these two packages. We also include some description of the WAM Cycle 4 (ST3) input source term formulation (Janssen, 1991), on which the ST4 input term is based, even though the ST3 package

was not tested in this study.

In addition to the input and dissipation terms, the other main control on deep-water wave transformation is provided by weakly nonlinear four-wave interactions (Hasselmann, 1962). Unfortunately, acceptable run time requirements for multiyear simulations over extensive domains still preclude using a near-exact computation of these terms, such as the Webb, Resio, Tracy method (Webb, 1978;Tracy and Resio, 1982) that is available in spectral models including WW3 (van Vledder et al.,

2000). Instead we use the much-simplified form of the Discrete Interaction Approximation (Hasselmann et al., 1985), treating its proportionality constant $C$ as a tunable parameter.

Common to both optimisations, sea ice obstruction was turned on (FLAGTR=4) with non-default values for the critical sea ice concentrations $\epsilon_{c,0}$ and $\epsilon_{c,n}$ between which wave obstruction by ice varies between zero and total blocking: these were set to 0.25 and 0.75, respectively. All other available parameters beyond the input and dissipation terms were left with default

settings, noting that shallow water processes, while activated, are not expected to have more than a negligible and localised influence on model outputs in a global simulation at 1° resolution.

For initial testing, in which two sets (ST2 and ST4) of optimisation parameters were compared, we used a one month (January 1997) spinup to a three month calibration period (February – April 1997). The selection of the calibration period from the full extent of the satellite record was arbitrary.

Relevant parameters used in the two calibrations are listed in Table 2 and Table 3, respectively, which refer to the parameter names as defined (more completely than we do here) in the WW3 user manual (Tolman, 2014), and as specified in namelist inputs to the model. These tables include the initial values of the parameters, the range over which they were allowed to vary, and the final optimised values. Other parameters not listed were kept fixed. A particular example was the input wind vertical level $z_r$ (ST2) $= z_u$ (ST4) $= 10$ m which is a property of the input data set, hence not appropriate to adjust. Others were left

fixed after an initial test confirmed that they had negligible influence on the objective function, leaving 13 adjustable parameters for ST2 and 17 for ST4.

The selection of which parameters to tune, and the range over which they are allowed to vary, is an area where some (partly subjective) judgement is still required, based on some familiarity with the relevant model parameterisations. In this case, parameter ranges were chosen to be physically realistic, and to cover the range of parameter choices used in previous studies reported in the literature.

### 3.3 Optimisation settings

We elected to primarily use the BOBYQA optimisation algorithm (Powell, 2009) for this study. Given that we expected WW3 to be already reasonably well-tuned for a global simulation such as our test case, we wished to use a local optimisation algorithm that could reach a solution to a problem with 10-20 variables in as few iterations as possible. Of the algorithms available in NLopt that were included in the intercomparison study of Rios and Sahinidis (2012), BOBYQA was found to be the most suitable in that respect. In particular it allows for concurrent model runs in the early stages of the optimisation process. Both optimisations were stopped when either the absolute change in (normalised) parameter values was less than 0.0001, or the relative change in the objective function was less than 0.0001. Less stringent conditions were initially used, but the ability of the optimisation suite to be restarted with revised stopping criteria was invoked to extend the optimisation.

These first two tests used a local optimisation method on the assumption that the respective default parameter sets are near-optimal, or at least within the "basin of attraction" of the optimal solution. In order to test this assumption, two further approaches can be considered. The first choice would be to use a truly global optimisation algorithm to explore the selected parameter space as thoroughly as possible. This approach may be expected to require a number if iterations in the thousands, which is rather challenging given typical model run times, especially as global methods do not generally allow for parallel iterations.

A simpler approach is to still use a local algorithm, but initialise it at a range of different starting points. This was the approach we took for our next set of tests, restricted to the ST4 case, in which the initial value of each parameter was selected at random with uniform probability distribution over its allowed range. Five randomised tests were done, along with a control optimisation starting from the default parameter set used previously. For these tests we made some further simplifications in the interests of computational speed, running the hindcast for only one month (February 1997), and initialising all simulations from a common initial condition, spun up over one month with the default parameter set. Both simplifications detract from how applicable the resulting parameter sets would be for hindcast applications, but can be justified in allowing a more extensive examination of parameter space with a given computational resource. A slightly reduced set of ST4 parameters was optimised, omitting $C_{ds}^{BCK}$, $C_{ds}^{HCK}$ and $s_B$. The initial and final values of these parameters from each of the tests are listed in Table 4 and Table 5, respectively. The allowed range of each of the adjustable parameters was the same as in the previous simulations, as listed in Table 3, while both stopping criteria were relaxed to a value of 0.005.

Despite the expected high computational demands, we next attempted an optimisation using the global evolutionary algorithm ESCH of da Silva Santos et al. (2010). This was initialised from the default parameter values, and used the same one month hindcast, parameter ranges and stopping criteria as described above.

Following these test simulations, the ST4 parameterisation was chosen for a final calibration, carried out over a 12 month period (January – December 1997) following a one-month spinup (December 1996). This calibration was finally terminated with both stopping criteria set to a value of 0.0001. This was a somewhat arbitrary choice made to observe the evolution of the solution. For practical applications the choice of stopping criteria should take into account the sensitivity of the objective function to measurement error in the data used for the calibration, to avoid unnecessary 'over-tuning' of the model.

The full hindcast, from January 1979 through December 2016 was then run using the optimised parameter set. Comparisons with altimeter data were made for each month from August 1991 onward.

Each WW3 simulation was run on 64 processors on a single core of either an IBM Power6 or a Cray XC50 machine. Other processing tasks within the suites were run on single processors. The resulting hindcast simulations required an average of approximately 25 minutes of wall clock time to complete each month of simulation.

## 4 Results

### 4.1 Local optimisation of 3 month hindcasts with ST2 and ST4 source terms

The BOBYQA algorithm develops a quadratic model of the objective function. To do so, the first iteration evaluates the objective function at the initial point, then perturbs each component in turn by a positive increment, then by an equal negative increment (leaving all other components at the initial value). This can be seen for the ST2 optimisation in Figure 4, in which the bottom panel shows the sequence of (normalised) parameter values tested. With 13 adjustable parameters, this amounts to 27 iterations in this preliminary phase. As this sequence of parameter values is fixed, independent of the resulting objective function values, all of the first 27 iterations could have been run simultaneously as detailed above, if permitted by the queuing system. We, however, applied a limit of 7 parallel iterations in line with anticipated resource limitations.

The 3-month ST2 optimisation only required a further 7 iterations after this initial phase to reach a stopping criterion. The ST2 default parameter settings used as the starting point for optimisation resulted in an objective function value of 0.1901, which was reduced to 0.1424 in the optimisation process.

In the optimal configuration, none of the tunable parameters were at either of the limits of their imposed range, indicating that convergence to a true minimum (at least locally) had been reached. Most of the parameters were only slightly modified from their initial values: the largest changes were in parameters $b_0$ (reduced from 0.0003 to 0.0002059) and $b_1$ (0.47 to 0.2493), both influencing the low frequency dissipation term.

The ST4 3-month optimisation was initialised with the default settings from the TEST451 case reported by Ardhuin et al (2010), for which the objective function returned a value of 0.1427. Optimisation only managed to reduce this to 0.1419 (Figure 5), indicating that the default ST4 parameter set was already quite closely tuned for our case, having been selected by Ardhuin et al (2010) largely from broadly similar studies, i.e. global simulations (at 0.5° resolution) compared with altimeter records. Three of the parameters ended the optimisation at one end of their allowed range, in each case at the same value at which it was initialised. The 16[th] adjustable parameter ($s_B$) controls the assumed directional spread of the dissipation spectrum, and the fact that it remained at its upper limit suggests that the optimisation may be improved by assuming the dissipation spectrum to have a narrower directional distribution than anticipated. On the other hand, parameters 14 ($C_{ds}^{BCK}$) and 15 ($C_{ds}^{HCK}$) are associated with an alternative breaking formulation proposed by Filipot and Ardhuin (2012), who chose values $C_{ds}^{BCK} = 0.185$ and $C_{ds}^{HCK} = 1.5$ (and correspondingly, turned off the default saturation-based dissipation term by setting $C_{ds}^{sat} = 0$) whereas this term is turned off in the ST4 default, hence both were initially set to zero. On the face of it, one might think that the optimisation algorithm would have been free to explore solutions with positive values of these parameters, resulting in an optimal 'hybrid' total dissipation term. In fact the way the dissipation algorithm is coded, this form of the dissipation term is not computed at all in the event that $C_{ds}^{BCK} = 0.0$, which would have been the case when the BOBYQA algorithm explored sensitivity to $C_{ds}^{HCK}$ in the initial stages. This means that our choice of initial values may have spuriously caused the BOBYQA algorithm to underestimate sensitivity to $C_{ds}^{HCK}$, and may have missed a distinct second local minimum (approximately corresponding to the parameter settings of Filipot and Ardhuin (2012)).

### 4.2 Tests with local optimisation with randomised initial parameter sets, and global optimisation

The next set of five tests compared results of the local BOBYQA algorithm starting from different parameter sets chosen at random within the allowed ranges (Table 4). The resulting final parameter sets, listed in Table 5, show that each test located a different minimum. This indicates that there are multiple local minima for the error metric in our chosen parameter ranges, in

addition to the local minimum derived from the default parameters. The corresponding values of the error metric were all slightly higher than the value (0.1454) obtained from the baseline optimisation starting from the default parameter set, although much reduced from their initial values (Table 4). Although none of those additional local minima found so far has replaced the baseline set as a candidate for a global optimum, this gives no guarantee that this would not be the case after a more thorough search.

The attempted global optimisation (using the ESCH algorithm) of the same hindcast, had not converged to within the chosen tolerances after 800 cycles. However, in the course of its operation it did identify over 30 parameter sets with slightly lower error metric than the minimum value (0.1450) obtained in the corresponding baseline local optimisation. The lowest value within 800 iterations was 0.1441, and the corresponding parameter values are included in Table 5. This supports our suspicion that a local optimisation algorithm cannot be relied upon to identify the global optimum for this hindcast problem. On the other hand, the very small decrease in the error metric obtained from this wider search does not give strong justification for making a significant change in parameters from near their default values. We need to bear in mind that the optimisation problem we have addressed in this set of tests (i.e. minimising RMS errors in significant wave height from a one month partial hindcast) is not quite the same as optimising this measure over a more representative period.

### 4.3 Local optimisation of 12 month hindcast with ST4 source terms

In the final 12 month ST4 optimisation, two additional parameters were allowed to vary that were fixed in the 3-month optimisation, bringing the number of adjustable parameters to 19. These were the critical sea ice concentration parameters $\epsilon_{c,0}$ and $\epsilon_{c,n}$ between which wave obstruction by ice varies between zero and total blocking: these had been fixed at 0.25 and 0.75, respectively, in the 3 month optimisations. Otherwise, the initial parameters (Table 4) again corresponded to the ST4 defaults, which in this case produced an error metric of 0.1436. At the termination after 89 iterations (with the more stringent stopping criteria), this had decreased to 0.1431.

Most of the resulting optimised parameters were close to the values obtained from the 3-month optimisation (Table 3). An exception was the 11th adjustable parameter, $C_{turb}$ , scaling the strength of the turbulent contribution to dissipation, which finished the 3-month optimisation at 0.41298, but at 0.0 (the lower bound) in the 12 month simulations.

For this longer optimisation, we have additionally computed a measure of the sensitivity of the objective function, using the initial phase of the BOBYQA iterations to estimate the change in the (un-normalised) parameter required to produce a 0.1% change in the objective function. This is listed as "Delta" in the seventh column of Table 3, and provides a measure, at least in relative terms, of the bounds within which each parameter value has been determined.

The full hindcast, run from 1979 to 2016, could be compared with satellite data from August 1991 onward. The resulting bias in significant wave height, averaged over the August 1991 – December 2016 comparison period, is shown in Figure 6. Positive biases are obtained in latitudes south of 45°S, particularly south of Australia and in the South Atlantic. This is also seen in the vicinity of some island groups (notably French Polynesia, Micronesia, the Maldives, Aleutians, Carribean, Azores), which may be indicative of insufficient sub-grid scale obstruction. On the other hand, negative biases are seen near the western sides of major ocean basins, and in the "swell shadow" to the northeast of New Zealand. A similar pattern is seen in the results reported by Ardhuin et al (2010) for their TEST441 case (their Figure 9).

Normalised root-mean-square error (i.e. RMSE error divided by the observed mean) from the same comparison, again averaged over the period August 1991 – December 2016, is shown in Figure 7. Note that the objective function for our optimisation used this measure, spatially averaged over ocean waters between 61°S and 61°N. For the majority of the ocean surface, this lies in the range 0.08 – 0.14, but with higher values near some island chains and the western boundaries of ocean basins. Again, similar results were reported by Ardhuin et al (2010).

**5 Discussion**

In their review of methods used to tune Numerical Weather Prediction and climate models, Hourdin et al. (2017) observe that with the number and complexity of parameterisations to consider, the task of tuning these parameters was for a long time largely left to "expert judgement", and that objective methods have made a more recent appearance than in the statistical, engineering, and computing fields. The method we have presented here, along with the approaches of Severijns and Hazeleger (2005), Tett et al. (2013), Roach et al. (2017) described in the introduction, perform model tuning through the relatively direct approach of defining and minimising a cost function. Our method has the advantage of employing a tool (Cylc) that is already commonly used to control complex workflows for weather forecasting and climate modelling systems, to optimize the parameters of such a system under its control, in a way that is simple to implement, and flexible in choice of optimisation algorithm.

We have shown this to be a practical method for optimising 10-20 parameters in a model application of sufficient complexity to require several hours per simulation in a parallel processing computing environment. For applications that are yet more time-consuming, it is becoming increasingly common (Bellprat et al., 2012;Wang et al., 2014;Duan et al., 2017) to first build a surrogate model to provide a statistical emulator for the actual model, and then apply an optimisation algorithm to the surrogate model. Such multi-stage model optimisation frameworks are beyond the scope of this paper, but the flexibility of our approach could potentially bring benefits to them as well. For example, it may be worth considering a hybrid approach of using a surrogate model to quantify the role of the full set of model parameters and perform an initial global optimisation, before switching to a method such as ours for a final refinement using the original model directly.

In our study we have largely restricted our attention to one local optimisation algorithm (BOBYQA), but our initial results suggest the need in some circumstances to apply a more global method. This is not difficult to do in principle, with multiple algorithms, both global and local, implemented in Cyclops. However, the generally higher computational demands of a global algorithm put a limit on such applications. In this study we have only been able to undertake a preliminary exploration of the wider parameter space of our single chosen test case. This did however illustrate that the possibility of multiple alternative local minima must be considered.

As we have seen, there remains a need for care with the choices of which parameters to attempt to optimise, and what bounds to set on their values. Most optimisation algorithms are intended for continuously variable parameters, and may rely on the objective function having a continuous dependence on these parameters. In many cases it is clear which parameters fall into this category, as opposed to discrete valued options. But in some cases, model code may make binary choices based on real parameters lying within discrete ranges, which may break this assumption. Hence the Cyclops optimisation suite is best employed in conjunction with a good understanding of the role each parameter plays in the model, and the interplay between them.

It is also important to be aware of the role played by the design of the error metric, which may make it sensitive to some parameters and insensitive to others. One should be wary of accepting a large change in these insensitive parameters to achieve a tiny improvement in the chosen error metric, when the resulting model could then perform poorly against other relevant criteria. In the particular wave modelling case we have investigated, our approach would not be sufficient on its own to identify suitable values of the large set of WW3 parameters without guidance from previous studies.

Tett et al. (2017) point out that the inherently chaotic nature of the climate system means that a certain level of noise is introduced into evaluations of an atmospheric model simulation, which can cause problems in evaluating the termination criteria. They describe a procedure to rerun a simulation that had nominally satisfied the prescribed convergence criteria, with randomised perturbations before determining whether or not to terminate. Unlike the atmosphere, ocean surface waves are an essentially dissipative system, and perturbations introduced in the initial conditions and forcing will tend to diminish, rather than grow, with time. As a result, noise in the objective function was not so relevant for our wave hindcast application as for

atmospheric models, but may need to be addressed in to systems with an underlying chaotic nature, possibly through implementing similar measures to those of Tett et al. (2017) into Cyclops.

Similarly, the dissipative nature of ocean waves means that a cost function based on a spatial average of the (temporal) RMSE of model-data comparisons will not be subject to the level of chaotic variability seen in similar measures for atmospheric

models. Small scale variability in wave model output is therefore more likely to be genuinely sensitive to parameter variation. In that case it is worth capturing such variability in the cost function, whereas for a chaotic system it may be wiser to average out such variability before evaluating the cost function.

**Conclusions**

The Cyclops Cylc-based optimisation suite offers a flexible tool for tuning the parameters of any modelling system that has

been implemented to run under the Cylc workflow engine. Minimal customisation of the modelling system is required beyond providing tasks to input and apply model parameter values in a simple namelist format, and output the value of the scalar error metric that is to be minimised. This then allows any of 16 optimisation algorithms (from the NLopt toolbox) to be applied to the optimisation. This optimisation suite is expected to be especially applicable to operational forecasting systems, where minimal re-configuration is required between "tuning" and "operational/production" versions of the forecast suite.

Results of the initial test case we have investigated, a global hindcast using a spectral wave model forced by ERA-Interim input fields, illustrate that the method is applicable to a modelling system of moderate complexity, both in terms of the number of parameters to tune, and the computational resources required, at least for the purposes of local optimisation to fine tune a model that already has a more-or-less well developed initial parameter set from previous studies. Investigations of systems that require a more global tuning approach, or are more computationally demanding remain for future work.

**Code availability**

Cyclops-v1.0 has been published through Zenodo (https://doi.org/10.5281/zenodo.837907) under a Creative Commons Attribution Share-Alike 4.0 licence.

Cylc is available from GitHub (https://cylc.github.io/cylc/) and Zenodo (https://zenodo.org/badge/latestdoi/1836229) under the GPLv3 licence.

**Appendix A: Handling concurrent simulations through dependencies**

An alternative way to allow for concurrent simulations involves modifying the simple Cylc suite described above to have several versions of the "Optimise" task. Now "Opt_*m*" runs the optimisation algorithm when there are *m* active model simulations still running, with *m* ranging from 0 to a set maximum *M-1*, where *M* is the maximum number of concurrent cycles we chose to allow. There are a more complex set of dependencies to ensure that this is the case. In particular, there is a condition

$$\text{Table}[-P(m+1)] => \textbf{Opt\_m}$$

to ensure that the lookup table has been updated with the results of all completed (i.e. inactive) cycles. If that is the case, the optimisation code will be run to determine if a new model simulation can be launched while those *m* tasks are active. If not, the suite will wait until one of the active model runs completes, and try again with "Opt_*m-1*", and so forth.

The dependency diagram for the case in which up to three concurrent simulations are allowed (i.e. *M* = 3) is illustrated in Figure 8. Assume, for example, that we are still well short of convergence, and that the optimisation algorithm is such that the next parameter set tested depends on all previous results. Then "Opt_2" and "Opt_1" will always give a "wait" message, and "Opt_0" will be needed on each cycle. This effectively produces the same behaviour as in Figure 1, with each cycle waiting for the immediately preceding cycle to complete before "Opt_0" can start, leading to a new model run. If, on the other hand, the algorithm never depends on the results of the previous two (active) calculations, "Opt_2" will always give a "next" message. This removes the "Opt_1 and Opt_0" tasks (and any dependencies upon them), leading to the "Model" task being called for cycle *N* as soon as the cycle *N-3* model run has completed and updated the lookup table, even if the cycle *N-2* and *N-1* "Model" tasks are still running.

**Appendix B: WW3 source term parameterisations**

**B.1 Tolman and Chalikov input + dissipation source term package**

The input source term is defined as

$$S_{in}(k, \theta) = \sigma\beta N(k, \theta) \tag{B1}$$

where $\beta$ is a non-dimensional wind-wave interaction parameter, which has a parameterised dependence on wind speed and direction, through boundary layer properties influenced by the wave spectrum. These dependencies are, however, fully determined with no user-adjustable terms, so we omit the details here.

This input term was, however, adjusted by Tolman (2002) following a global test case to ameliorate an excessive dissipation of swell in weak or opposing winds, in which cases $\beta$ can be negative. This is done by applying, when $\beta$ is negative, a swell filtering scaling factor with a constant value $X_s$ for frequencies below $0.6f_p$ (where $f_p$ is the peak frequency), scaling linearly up to 1 at $0.8f_p$, with higher frequencies unmodified.

The same study also led to the introduction of a correction for the effects of atmospheric stability on wave growth identified by Kahma and Calkoen (1992) by replacing the wind speed $u$ with an effective wind speed $u_e$, with

$$\left(\frac{u_e}{u}\right)^2 = 1 + c_1 \tanh(\max(0, f_1\{\mathcal{ST} - \mathcal{ST}_0\})) + c_2 \tanh\left(\max\left(0, f_1\frac{c_1}{c_2}\{\mathcal{ST} - \mathcal{ST}_0\}\right)\right) \tag{B2}$$

where $\mathcal{ST}$ is a bulk stability parameter

$$\mathcal{ST} = \frac{hg}{u_h^2}\frac{T_a - T_s}{T_0} \tag{B3}$$

in terms of air, sea and reference temperatures $T_a$, $T_s$ and $T_0$, respectively, and $u_h$ the wind speed at reference height $h = 10$ m, with $g$ the gravitational acceleration. As air and sea surface temperature fields are available from the ERA-Interim dataset, it was possible to apply this parametrisation, treating $c_0$, $c_1$, $c_2$, $f_1$ and $\mathcal{ST}_0$ as adjustable dimensionless parameters.

The dissipation term consists of a dominant low-frequency constituent, with an empirical frequency dependence parameterised by constants $b_0$, $b_1$, $\phi_{min}$ and a high-frequency term, parameterised by constants $a_0$, $a_1$, $a_2$, the details of which we leave for the WW3 manual (Tolman, 2014) and original references therein.

**B.2 WAM Cycle 4 source term package**

The input source term implemented in WAM Cycle4 by Janssen (1982) was based on the wave growth theory of Miles (1957). The starting point is the assumption the wind speed $U$ has a logarithmic profile, so that if the wind fields input to the model are specified at elevation $z_u$ , then

$$U(z_u) = \frac{u_*}{\kappa} \log\left(\frac{z_u}{z_1}\right) \tag{B4}$$

where $u_*$ is the friction velocity, defined by the total wind stress $\tau = u_*^2$, $\kappa$ is von Karman's constant, and $z_1$ is a roughness length modified by wave conditions:

$$z_1 = \frac{z_0}{\sqrt{1 - \tau_w/\tau}} \tag{B5}$$

in which $\tau_w$ is the magnitude of the wave-supported stress, while

$$z_0 = \alpha_0\,\tau/g \tag{B6}$$

with $\alpha_0$ a tunable dimensionless parameter.

The wave-supported stress can be equated to the rate of momentum transfer between wind and waves:

$$\vec{\tau}_w = \int dk\,d\theta\,\frac{\vec{k}}{C}S_{in}(k,\theta) \tag{B7}$$

where $c$ is the wave phase velocity

The WAM Cycle 4 input source term is then given by

$$S_{in}(k,\theta) = \frac{\rho_a}{\rho_w}\frac{\beta_{max}}{\kappa^2}e^Z Z^4\left(\frac{u_*}{C} + z_\alpha\right)^2 [\max(\cos(\theta - \theta_u), 0)]^{p_{in}}\sigma N(k,\theta) + S_{out}(k,\theta) \tag{B8}$$

with

$$Z = \log(kz_1) + \frac{\kappa}{\cos(\theta - \theta_u)\left(\frac{u_*}{C} + z_\alpha\right)} \tag{B9}$$

In these terms $\rho_a$ and $\rho_w$ are the densities of air and water, $\beta_{max}$ is a dimensionless constant, $z_\alpha$ is a wave age tuning parameter and $p_{in}$ is a parameter controlling the directional dependence relative to the wind direction $\theta_u$.

The inter-dependence of $\tau_w$ and $S_{in}$ expressed in (B7) and (B8) creates an implicit functional dependence of $u_*$ on $U$ and $\tau_w/\tau$. In practice, this dependence can be tabulated, using the resolved model spectrum for the low-frequency ($k < k_{max}$) part of (B7), above which a $f^{-5}$ diagnostic tail is assumed.

The $S_{out}$ term represents a linear damping of swells, in the form (Bidlot, 2012):

$$S_{out}(k,\theta) = 2s_1\kappa\frac{\rho_a}{\rho_w}\left(\frac{u_*}{C}\right)^2\left[\cos(\theta - \theta_u) - \frac{\kappa C}{u^*\log(kz_0)}\right]\sigma N(k,\theta) \tag{B10}$$

with $s_1$ set to 1(0) to turn on(off) the damping.

Dissipation is represented in the form

$$S_{ds}(k,\theta) = C_{ds}\bar{\alpha}^2\bar{\sigma}\left[\delta_1\frac{k}{\bar{k}} + \delta_2\left(\frac{k}{\bar{k}}\right)^2\right]N(k,\theta) \tag{B11}$$

where $C_{ds}$ is a dimensionless constant, and $\delta_1$ and $\delta_2$ are weighting parameters. These take values $C_{ds} = -1.33$, $\delta_1 = 0.5$ and $\delta_2 = 0.5$ in the ECMWF implementation of WAM as reported by Bidlot (2012), but are adjustable within WW3. Mean wavelength and frequency are defined as

$$\bar{k} = \left[\frac{\int k^p N(k,\theta)d\vec{k}}{\int N(k,\theta)d\vec{k}}\right]^{1/p} \tag{B12}$$

and

$$\bar{\sigma} = \left[\frac{\int \sigma^p N(k,\theta)d\vec{k}}{\int N(k,\theta)d\vec{k}}\right]^{1/p} \tag{B13}$$

with $p = 0.5$ and $p = 1$ being the respective WAM defaults (Bidlot, 2012) while mean steepness is

$$\bar{\alpha} = E\bar{k}^2 \tag{B14}$$

## B.3 Ardhuin (2010) source term package

This package introduces a saturation-based dissipation term. In order to accommodate this, the WAM Cycle 4 input source function is modified by replacing $u_*$ in (B8) with a frequency-dependent form

$$\left(u_*'(k)\right)^2 = \left| u_*^2 - |s_u| \left| \int_0^k dk' \int d\theta \frac{\vec{k'}}{C} S_{in}(k',\theta) \right| \right| \tag{B15}$$

in which $s_u \approx 1$ is a sheltering coefficient, to allow for balance with a saturation-based dissipation term. Also, a limit can be placed on the roughness length $z_0$, replacing (B6) with

$$z_0 = \min(\alpha_0 \tau/g, z_{0,\max}) \tag{B16}$$

The swell dissipation parameterisation of Ardhuin et al. (2009) is used, consisting of terms

$$S_{out,visc}(k,\theta) = -s_5 \frac{\rho_a}{\rho_w}\left[2k\sqrt{2\nu_a\sigma}\right]N(k,\theta) \tag{B17}$$

and

$$S_{out,turb}(k,\theta) = -\frac{\rho_a}{\rho_w}\left[16f_e\sigma^2 u_{orb,s}/g\right]N(k,\theta) \tag{B18}$$

due to effects of the viscous and turbulent boundary layers respectively. The latter depends on the significant surface orbital velocity

$$u_{orb,s} = 2\left[\int dk d\theta\, \sigma^3 N(k,\theta)\right]^{1/2} \tag{B19}$$

while $\nu_a$ is air viscosity and $s_5$ is a tunable coefficient of order 1. The two terms are combined in weighted form

$$S_{out}(k,\theta) = r_- S_{out,vis}(k,\theta) + r_+ S_{out,turb}(k,\theta) \tag{B20}$$

with weights

$$r_\pm = 0.5(1 \pm \tanh((Re - Re_c')/s_7)) \tag{B21}$$

depending on a modified air-sea boundary layer Reynolds number

$$Re = 2u_{orb,s}H_s/\nu_a \tag{B22}$$

which is taken to have a threshold value depending on significant wave height:

$$Re_c' = Re_c(4m/H_s)^{1-s_6} \tag{B23}$$

The turbulent dissipation term is parameterised to depend on wind speed and direction:

$$f_e = s_1 f_{e,GM} + [|s_3| + s_2\cos(\theta - \theta_u)]u_*/u_{orb} \tag{B24}$$

based on the friction factor $f_{e,GM}$ from the Grant and Madsen (1979) theory of oscillatory boundary layer flow over a rough surface.

The dissipation term is based on the saturation of the wave spectrum, and takes the form

$$S_{ds}(k,\theta) = \sigma \frac{C_{ds}^{sat}}{B_r^2}[\delta_d \max(B(k) - B_r, 0)^2 + (1 - \delta_d)\max(B'(k,\theta) - B_r, 0)^2]N(k,\theta)$$
$$+ S_{bk,cu}(k,\theta) + S_{turb}(k,\theta)$$

(B25)

where the dissipation spectrum is integrated over a limited direction range, i.e.

$$B'(k,\theta) = \int_{\theta - \Delta_\theta}^{\theta + \Delta_\theta} \sigma k^3 \cos^{sB}(\theta - \theta')N(k,\theta)d\theta'$$

(B26)

and

$$B(k) = \max(B'(k,\theta), \theta \in [0, 2\pi])$$

(B27)

The cumulative breaking term, associated with large scale breakers overtaking short waves, is

$$S_{bk,cu}(k,\theta) = \frac{-14.2 C_{cu}}{\pi^2} N(k,\theta) \int_0^{r_{cu}^2 k} dk' \int_0^{2\pi} d\theta' \max\left\{ \sqrt{B(f',\theta')} - \sqrt{B_r}, 0 \right\}^2$$

(B28)

Where $r_{cu} = 0.5$ and $C_{cu}$ is a tuning coefficient.

The turbulent dissipation term is

$$S_{turb}(k,\theta) = -2C_{turb}\sigma \cos(\theta_u - \theta)k\frac{\rho_a u_*^2}{g\rho_w}N(k,\theta)$$

(B29)

An alternative breaking formulation (Filipot and Ardhuin, 2012) based on a bore model uses a dissipation rate per unit crest length of

$$\epsilon_{CK} = \frac{1}{4}\rho_w g \left[\frac{C_{ds}^{BCK}H}{\tanh(kh)^{C_{ds}^{HCK}}}\right]^3 \sqrt{\frac{gk}{\tanh(kh)}}$$

(B30)

**Competing interests**

The authors declare that they have no conflict of interest.

**Author contribution**

Richard Gorman developed the Cyclops optimisation suite, carried out all simulations, and prepared the manuscript. Hilary Oliver leads the development of Cylc, assisted with the design of the optimisation suite, and contributed to the content of the manuscript.

**Acknowledgements**

This work was supported by Strategic Science Investment Funding provided to NIWA from the New Zealand Ministry of Business, Innovation and Employment (MBIE). The authors wish to acknowledge the contribution of the New Zealand eScience Infrastructure (NeSI: http://www.nesi.org.nz) to the results of this research. New Zealand's national compute and analytics services and team are supported by NeSI and funded jointly by NeSI's collaborator institutions and through MBIE.

We appreciate feedback on the manuscript from Lettie Roach.

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

**Table 1 Derivative-Free Optimisation algorithms from the NLopt toolbox supported in the Cyclops optimisation suite**

| |
|---|
| **Global:** |
| DIRECT: Dividing RECTangles (Jones et al., 1993) |
| DIRECT-L: Dividing RECTangles, locally optimised (Gablonsky and Kelley, 2001) |
| DIRECT-L-RAND: a slightly randomised variant of DIRECT-L (Johnson, 2014) |
| CRS: Controlled Random Search (Hendrix et al., 2001) |
| CRS2: Controlled Random Search (Price, 1983) |
| CRS2-LM: Controlled Random Search with Local Mutation (Kaelo and Ali, 2006) |
| MLSL: Multi-Level Single-Linkage (Rinnooy Kan and G. T. Timmer, 1987) |
| ISRES: Improved Stochastic Ranking Evolution Strategy (Runarsson and Yao, 2005) |
| ESCH: Evolutionary algorithm (da Silva Santos et al., 2010) |
| **Local:** |
| COBYLA: Constrained Optimization BY Linear Approximations (Powell, 1994) |
| BOBYQA: Bounded Optimization BY Quadratic Approximation (Powell, 2009) |
| NEWUOA: Unconstrained Optimization (Powell, 2004) |
| NEWUOA-BOUND: a bounded variant of NEWUOA (Johnson, 2014) |
| PRAXIS: Principal Axis (Brent, 1972) |
| Nelder-Mead Simplex (Nelder and Mead, 1965) |
| Sbplx: Nelder-Mead applied on a sequence of subspaces (Rowan, 1990) |

**Table 2. Parameters used to calibrate the simulation using the source term package of Tolman and Chalikov (1996), for February through April 1997. The first two columns list the parameter as defined in the WW3 v4.18 user manual (Tolman, 2014), and as specified in WW3 namelist input. The namelist groupings in bold correspond to parameterisations related to wind input (SIN2), dissipation (SDS2), nonlinear interactions (SNL1), and some "miscellaneous" parameters (MISC). Lower and upper bounds are specified for parameters adjusted during calibration, along with their final values, and the corresponding index $n$ of the normalised parameter vector, as used to label plots in Figure 4. Other parameters were fixed at the initial value.**

| Parameter | Code variable | Initial | Lower bound | Upper bound | Final | $n$ |
|---|---|---|---|---|---|---|
| | **SIN2:** | | | | | |
| $X_s$ | SWELLF | 0.1 | 0.0 | 1.0 | 0.1175 | 1 |
| $c_0$ | STABSH | 1.38 | 1.0 | 1.8 | 1.374 | 2 |
| $\mathcal{ST}_0$ | STABOF | -0.01 | -0.02 | -0.001 | -0.01031 | 3 |
| $c_1$ | CNEG | -0.01 | -0.02 | -0.001 | -0.01033 | 4 |
| $c_2$ | CPOS | 0.01 | 0.001 | 0.02 | 0.009666 | 5 |
| $-f_1$ | FNEG | 150.0 | 100.0 | 200.0 | 148.25 | 6 |
| | **SDS2:** | | | | | |
| $a_0$ | SDSA0 | 4.8 | 4.0 | 6.0 | 4.8045 | 7 |
| $a_1$ | SDSA1 | $1.7 \times 10^{-4}$ | $1.0 \times 10^{-4}$ | $5.0 \times 10^{-3}$ | $1.7023 \times 10^{-4}$ | 8 |
| $a_2$ | SDSA2 | 2.0 | 1.0 | 4.0 | 2.0120 | 9 |
| $b_0$ | SDSB0 | 0.3E-3 | -0.01 | 0.01 | 0.0002059 | 10 |
| $b_1$ | SDSB1 | 0.47 | 0.2 | 1.0 | 0.2494 | 11 |
| $\phi_{min}$ | PHIMIN | 0.003 | 0.002 | 0.005 | 0.002972 | 12 |
| | **SNL1:** | | | | | |
| $C$ | NLPROP | $2.5 \times 10^{-7}$ | $2.4 \times 10^{-7}$ | $2.8 \times 10^{-7}$ | $2.498 \times 10^{-7}$ | 13 |

**Table 3.** As for Table 2, but for parameters used to calibrate the simulation using the source term package of Ardhuin et al (2010), for February through April 1997. The namelist groupings in bold correspond to parameterisations related to wind input (SIN4), dissipation (SDS4), nonlinear interactions (SNL1), and some "miscellaneous" parameters (MISC). Lower and upper bounds are specified for parameters adjusted during calibration, along with their final values, and the corresponding index $n$ of the normalised parameter vector, as used to label plots in Figure 5.

| Parameter | Code variable | Initial | Lower bound | Upper bound | Final | $n$ |
|---|---|---|---|---|---|---|
| | **SIN4:** | | | | | |
| $\beta_{max}$ | BETAMAX | 1.52 | 1.0 | 2.0 | 1.5197 | 1 |
| $s_u$ | TAUWSHELTER | 1.0 | 0.0 | 1.5 | 0.9594 | 2 |
| $s_2$ | SWELLF | 0.8 | 0.5 | 1.2 | 0.8010 | 3 |
| $s_1$ | SWELLF2 | -0.018 | -0.03 | -0.01 | -0.01812 | 4 |
| $s_3$ | SWELLF3 | 0.015 | 0.01 | 0.02 | 0.01484 | 5 |
| $Re_c$ | SWELLF4 | $1.0 \times 10^5$ | $0.8 \times 10^5$ | $1.5 \times 10^5$ | $0.9973 \times 10^5$ | 6 |
| $s_5$ | SWELLF5 | 1.2 | 0.8 | 1.6 | 1.2078 | 7 |
| $s_7$ | SWELLF7 | $2.3 \times 10^5$ | 0.0 | $4.0 \times 10^5$ | $2.2600 \times 10^5$ | 8 |
| | **SDS4:** | | | | | |
| $C_{ds}^{sat}$ | SDSC2 | $-2.2 \times 10^{-5}$ | $-2.5 \times 10^{-5}$ | 0.0 | $-2.1506 \times 10^{-5}$ | 9 |
| $C_{cu}$ | SDSCUM | -0.40344 | -0.5 | 0.0 | -0.4020 | 10 |
| $C_{turb}$ | SDSC5 | 0.0 | 0.0 | 1.2 | 0.4168 | 11 |
| $\delta_d$ | SDSC6 | 0.3 | 0.0 | 1.0 | 0.2654 | 12 |
| $B_r$ | SDSBR | 0.0009 | 0.0008 | 0.0010 | 0.0009035 | 13 |
| $C_{ds}^{BCK}$ | SDSBCK | 0.0 | 0.0 | 0.2 | 0.0 | 14 |
| $C_{ds}^{HCK}$ | SDSHCK | 0.0 | 0.0 | 2.0 | 0.0933 | 15 |
| $s_B$ | SDSCOS | 2.0 | 0.0 | 2.0 | 2.0 | 16 |
| | **SNL1:** | | | | | |
| $C$ | NLPROP | $2.5 \times 10^{-7}$ | $2.4 \times 10^{-7}$ | $2.8 \times 10^{-7}$ | $2.510 \times 10^{-7}$ | 17 |

**Table 4. Initial parameters used to calibrate the simulations using the source term package of Ardhuin et al (2010), for Feb 1997, using randomised initial conditions (simulations 1-5). Simulation 0 is the control case, with default initial parameters.**

| Parameter | Code variable | Simulation number | | | | | |
|---|---|---|---|---|---|---|---|
| | | 0 | 1 | 2 | 3 | 4 | 5 |
| | **SIN4:** | | | | | | |
| $\beta_{max}$ | BETAMAX | 1.520 | 1.215 | 1.160 | 1.538 | 1.660 | 1.550 |
| $s_u$ | TAUWSHELTER | 1.000 | 0.244 | 1.281 | 1.381 | 0.996 | 0.950 |
| $s_2$ | SWELLF | 0.800 | 0.962 | 0.948 | 0.582 | 0.995 | 1.026 |
| $s_1$ | SWELLF2 | -0.018 | -0.022 | -0.012 | -0.026 | -0.0253 | -0.018 |
| $s_3$ | SWELLF3 | 0.015 | 0.016 | 0.014 | 0.0116 | 0.0131 | 0.0159 |
| $Re_c$ | SWELLF4 | $1.000 \times 10^5$ | $1.428 \times 10^5$ | $1.368 \times 10^5$ | $1.295 \times 10^5$ | $0.837 \times 10^5$ | $0.809 \times 10^5$ |
| $s_5$ | SWELLF5 | 1.200 | 1.100 | 1.411 | 1.589 | 1.290 | 1.290 |
| $s_7$ | SWELLF7 | $2.300 \times 10^5$ | $1.188 \times 10^5$ | $2.908 \times 10^5$ | $0.621 \times 10^5$ | $2.492 \times 10^5$ | $2.905 \times 10^5$ |
| | **SDS4:** | | | | | | |
| $C_{ds}^{sat}$ | SDSC2 | $-2.200 \times 10^{-5}$ | $-1.528 \times 10^{-5}$ | $-1.069 \times 10^{-5}$ | $-1.493 \times 10^{-5}$ | $-1.639 \times 10^{-5}$ | $-1.303 \times 10^{-5}$ |
| $C_{cu}$ | SDSCUM | -0.403 | -0.159 | -0.470 | -0.488 | -0.205 | -0.387 |
| $C_{turb}$ | SDSC5 | 0.000 | 1.116 | 1.074 | 1.025 | 0.476 | 0.882 |
| $\delta_d$ | SDSC6 | 0.300 | 0.957 | 0.596 | 0.947 | 0.855 | 0.583 |
| $B_r$ | SDSBR | $9.00 \times 10^{-4}$ | $9.13 \times 10^{-4}$ | $8.24 \times 10^{-4}$ | $8.14 \times 10^{-4}$ | $9.73 \times 10^{-4}$ | $8.39 \times 10^{-4}$ |
| | **SNL1:** | | | | | | |
| $C$ | NLPROP | $2.500 \times 10^7$ | $2.690 \times 10^7$ | $2.794 \times 10^7$ | $2.644 \times 10^7$ | $2.780 \times 10^7$ | $2.437 \times 10^7$ |
| | **Initial error score** | 0.1454 | 0.1685 | 0.2346 | 0.1722 | 0.2156 | 0.1677 |

**Table 5. Final values of parameters from simulations using the source term package of Ardhuin et al (2010), for Feb 1997, using BOBYQA with randomised initial conditions (simulations 1-5), and using ESCH with default initial parameters. Simulation 0 is the control case, using BOBYQA with default initial parameters.**

| Parameter | Code variable | Simulation number | | | | | | |
| --- | --- | --- | --- | --- | --- | --- | --- | --- |
| | | 0 | 1 | 2 | 3 | 4 | 5 | ESCH |
| | **SIN4:** | | | | | | | |
| $\beta_{max}$ | BETAMAX | 1.515 | 1.348 | 1.221 | 1.671 | 1.491 | 1.599 | 1.520 |
| $s_u$ | TAUWSHELTER | 0.950 | 0.244 | 1.275 | 1.385 | 1.035 | 0.953 | 0.898 |
| $s_2$ | SWELLF | 0.811 | 0.761 | 0.872 | 0.591 | 1.065 | 0.986 | 0.800 |
| $s_1$ | SWELLF2 | -0.0178 | -0.0256 | -0.0120 | -0.0148 | -0.0226 | -0.0248 | -0.018 |
| $s_3$ | SWELLF3 | 0.0149 | 0.0168 | 0.0134 | 0.0112 | 0.0150 | 0.0170 | 0.0150 |
| $Re_c$ | SWELLF4 | $0.996\times10^5$ | $1.428\times10^5$ | $1.376\times10^5$ | $1.339\times10^5$ | $0.837\times10^5$ | $0.809\times10^5$ | $1.198\times10^5$ |
| $s_5$ | SWELLF5 | 1.201 | 1.099 | 1.406 | 1.589 | 1.291 | 1.290 | 0.973 |
| $s_7$ | SWELLF7 | $2.30\times10^5$ | $1.19\times10^5$ | $2.84\times10^5$ | $0.64\times10^5$ | $2.47\times10^5$ | $2.89\times10^5$ | $2.42\times10^5$ |
| | **SDS4:** | | | | | | | |
| $C_{ds}^{sat}$ | SDSC2 | $-2.12\times10^{-5}$ | $-1.75\times10^{-5}$ | $-0.09\times10^{-5}$ | $-1.93\times10^{-5}$ | $-2.05\times10^{-5}$ | $-1.29\times10^{-5}$ | $-2.34\times10^{-5}$ |
| $C_{cu}$ | SDSCUM | -0.401 | -0.158 | -0.469 | -0.488 | -0.209 | -0.387 | -0.454 |
| $C_{turb}$ | SDSC5 | 0.386 | 1.116 | 1.067 | 1.027 | 0.526 | 0.831 | 0.567 |
| $\delta_d$ | SDSC6 | 0.246 | 0.957 | 0.560 | 0.940 | 0.860 | 0.585 | 0.043 |
| $B_r$ | SDSBR | $9.03\times10^{-4}$ | $9.19\times10^{-4}$ | $8.26\times10^{-4}$ | $8.20\times10^{-4}$ | $9.72\times10^{-4}$ | $8.38\times10^{-4}$ | $9.09\times10^{-4}$ |
| | **SNL1:** | | | | | | | |
| $C$ | NLPROP | $2.51\times10^7$ | $2.69\times10^7$ | $2.80\times10^7$ | $2.69\times10^7$ | $2.78\times10^7$ | $2.44\times10^7$ | $2.45\times10^7$ |
| | **Error score** | 0.1450 | 0.1479 | 0.1513 | 0.1515 | 0.1501 | 0.1500 | 0.1441 |
| | **Iterations** | 38 | 37 | 41 | 62 | 37 | 39 | 800+ (not converged) |

**Table 6. As for Table 3, but for parameters used to calibrate the simulation using the source term package of Ardhuin et al (2010), for Jan-Dec 1997. The "Delta" value in the seventh column is the estimated change in the (un-normalised) parameter required to produce a 0.1% change in the objective function.**

| Parameter | Code variable | Initial | Lower bound | Upper bound | Final | Delta | $n$ |
|---|---|---|---|---|---|---|---|
| | **SIN4:** | | | | | | |
| $\beta_{max}$ | BETAMAX | 1.52 | 1.0 | 2.0 | 1.5194 | 0.02498 | 1 |
| $s_u$ | TAUWSHELTER | 1.0 | 0.0 | 1.5 | 0.9339 | 0.2706 | 2 |
| $s_2$ | SWELLF | 0.8 | 0.5 | 1.2 | 0.8224 | 0.0206 | 3 |
| $s_1$ | SWELLF2 | -0.018 | -0.03 | -0.01 | -0.01721 | 0.00064 | 4 |
| $s_3$ | SWELLF3 | 0.015 | 0.01 | 0.02 | 0.01526 | 0.00042 | 5 |
| $Re_c$ | SWELLF4 | $1.0 \times 10^5$ | $0.8 \times 10^5$ | $1.5 \times 10^5$ | $0.9888 \times 10^5$ | $0.2328 \times 10^5$ | 6 |
| $s_5$ | SWELLF5 | 1.2 | 0.8 | 1.6 | 0.9360 | 0.3974 | 7 |
| $s_7$ | SWELLF7 | $2.3 \times 10^5$ | 0.0 | $4.0 \times 10^5$ | $2.2433 \times 10^5$ | $0.7911 \times 10^5$ | 8 |
| | **SDS4:** | | | | | | |
| $C_{ds}^{sat}$ | SDSC2 | $-2.2 \times 10^{-5}$ | $-2.5 \times 10^{-5}$ | 0.0 | $-2.1433 \times 10^{-5}$ | $0.0087 \times 10^{-5}$ | 9 |
| $C_{cu}$ | SDSCUM | -0.40344 | -0.5 | 0.0 | -0.40194 | 0.02145 | 10 |
| $C_{turb}$ | SDSC5 | 0.0 | 0.0 | 1.2 | 0.0 | - | 11 |
| $\delta_d$ | SDSC6 | 0.3 | 0.0 | 1.0 | 0.2736 | 0.0928 | 12 |
| $B_r$ | SDSBR | $9.0 \times 10^{-4}$ | $8.0 \times 10^{-4}$ | $10.0 \times 10^{-4}$ | $8.9788 \times 10^{-4}$ | $0.0951 \times 10^{-4}$ | 13 |
| $C_{ds}^{BCK}$ | SDSBCK | 0.0 | 0.0 | 0.2 | 0.0 | - | 14 |
| $C_{ds}^{HCK}$ | SDSHCK | 0.0 | 0.0 | 2.0 | 0.0 | - | 15 |
| $s_B$ | SDSCOS | 2.0 | 0.0 | 2.0 | 2.0 | 0.0757 | 16 |
| | **SNL1:** | | | | | | |
| $C$ | NLPROP | $2.5 \times 10^7$ | $2.4 \times 10^7$ | $2.8 \times 10^7$ | $2.5181 \times 10^7$ | $0.1191 \times 10^7$ | 17 |
| | **MISC:** | | | | | | |
| $\epsilon_{c,0}$ | CICE0 | 0.25 | 0.15 | 0.45 | 0.2413 | 0.1285 | 18 |
| $\epsilon_{c,n}$ | CICEN | 0.75 | 0.55 | 0.85 | 0.7521 | 0.2358 | 19 |

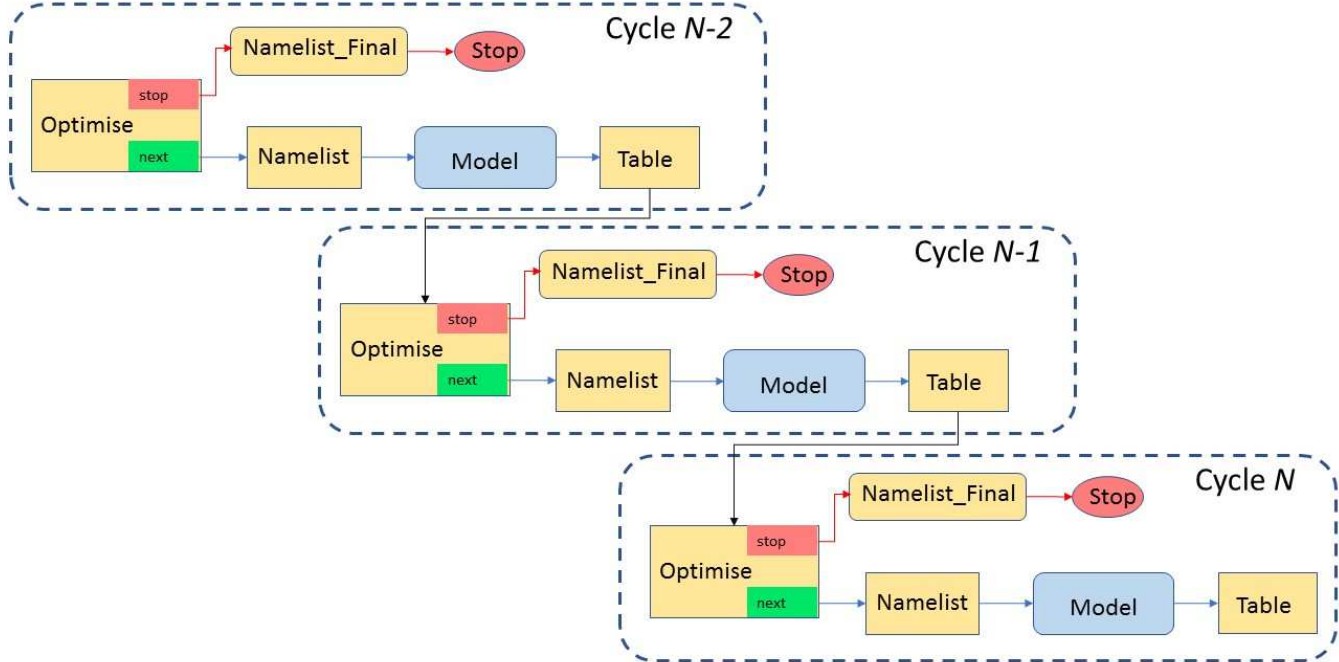

**Figure 1: Dependency graph for a version of the Cyclops optimisation suite in which no concurrent simulations are allowed, showing three successive cycles. Arrows represent dependency, in that a task at the head of an arrow depends on the task at the tail of the arrow meeting a specified condition (by default, this means completing successfully) before it can start.**

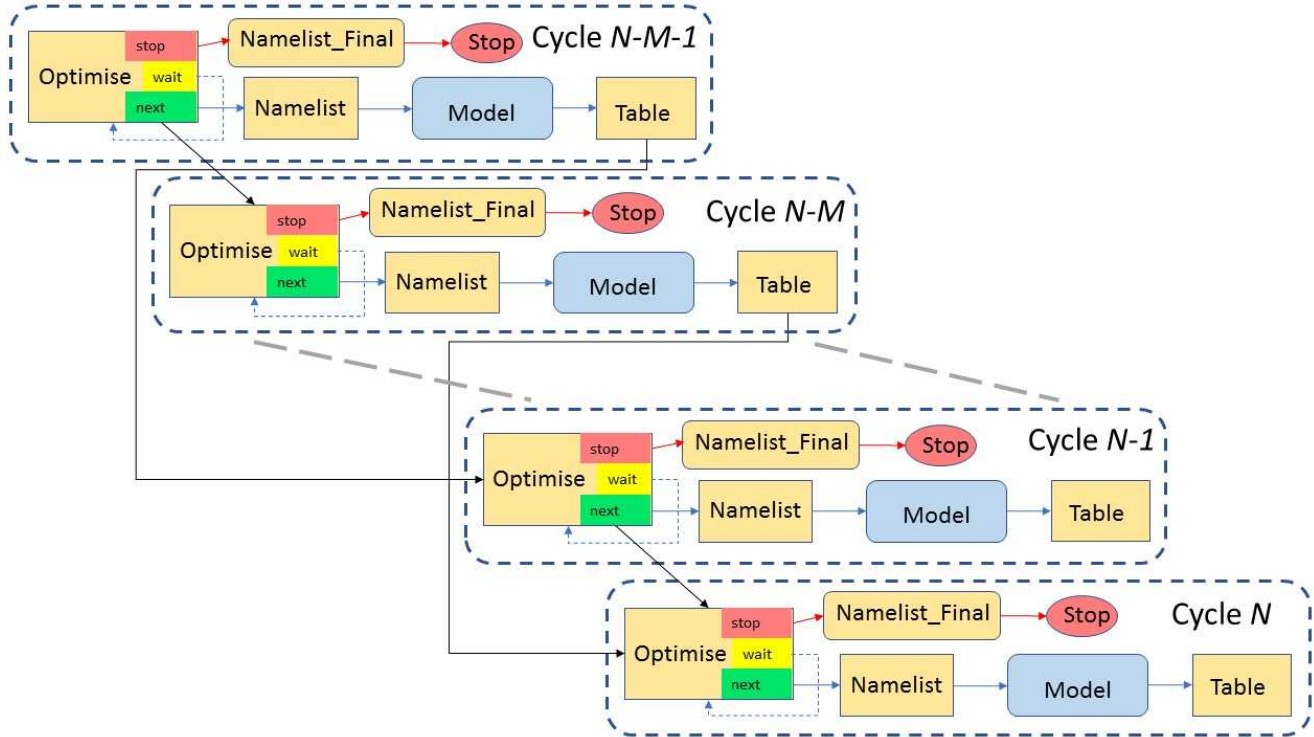

**Figure 2: Dependency graph for an implementation of the Cyclops optimisation suite in which up to *M* concurrent simulations are supported. Solid arrows represent dependency, in that a task at the head of an arrow depends on the task at the tail of the arrow meeting a specified condition (by default, this means completing successfully) before it can start. The dashed arrows represent a task retrying after a set interval. Only four cycles are shown, omitting tasks in intervening cycles, and their dependencies.**

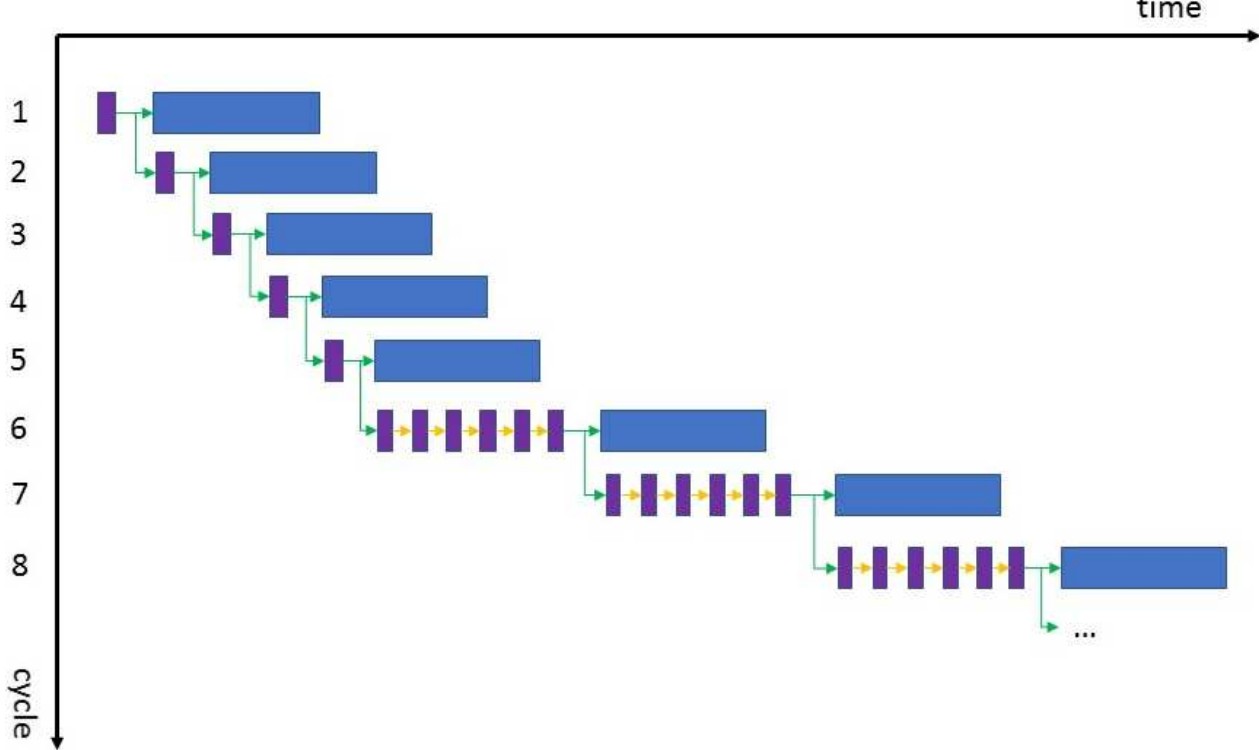

**Figure 3: Example of the cycling behaviour of an implementation of the Cyclops optimisation suite in which concurrent simulations are supported. Optimise tasks (purple boxes) which succeed trigger further tasks in the same cycle (blue boxes representing a sequence of Namelist, Model and Table tasks), and the Optimise task in the next cycle. Green arrows represent these dependencies on task success. Optimise tasks which fail to select a parameter vector independent of the result of active tasks retry (yellow arrows) at prescribed intervals until they succeed. The time axis is not to scale: Model tasks will typically have run times orders of magnitude longer than the run times of Optimise tasks. In this example, we suppose that the particular optimisation algorithm employed allows for up to five concurrent cycles during the initial stages.**

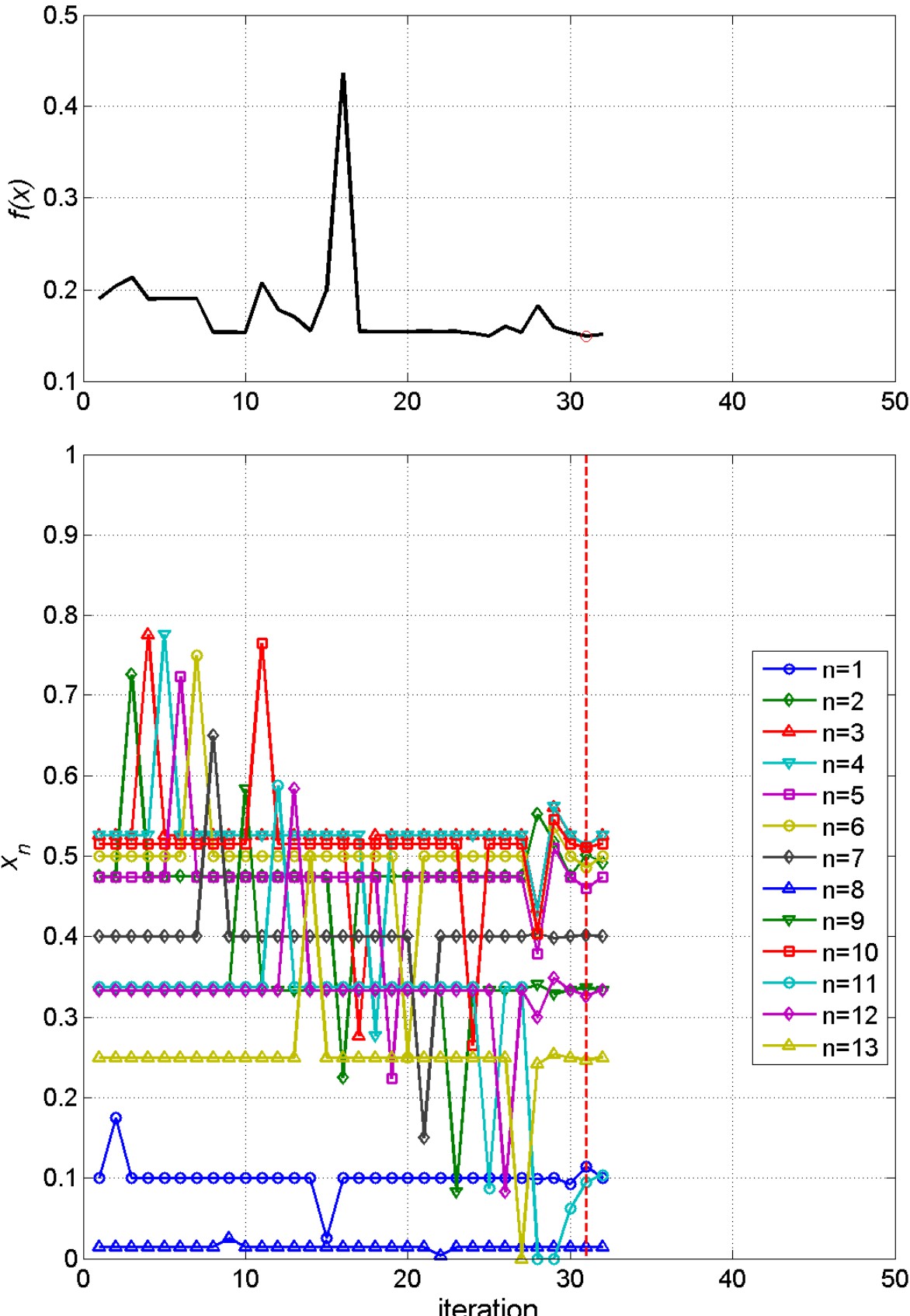

**Figure 4: Sequence of objective function values (top) and parameter vector components (bottom) at each iteration in the three month (February – April 1997) ST2 calibration. The red dashed line marks the optimal solution found.**

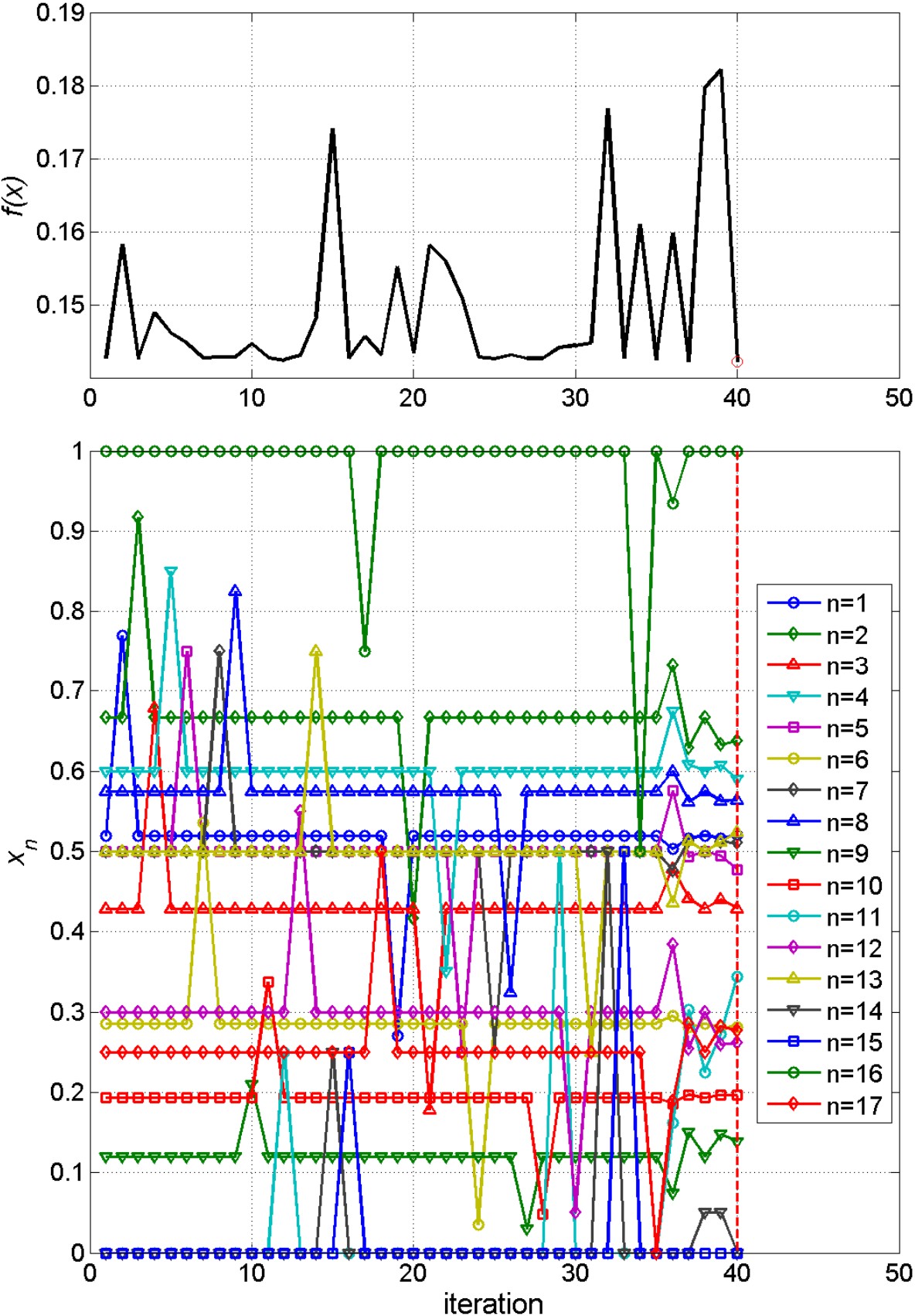

**Figure 5: Sequence of objective function values (top) and parameter vector components (bottom) at each iteration in the three month (February – April 1997) ST4 calibration. The red dashed line marks the optimal solution found.**

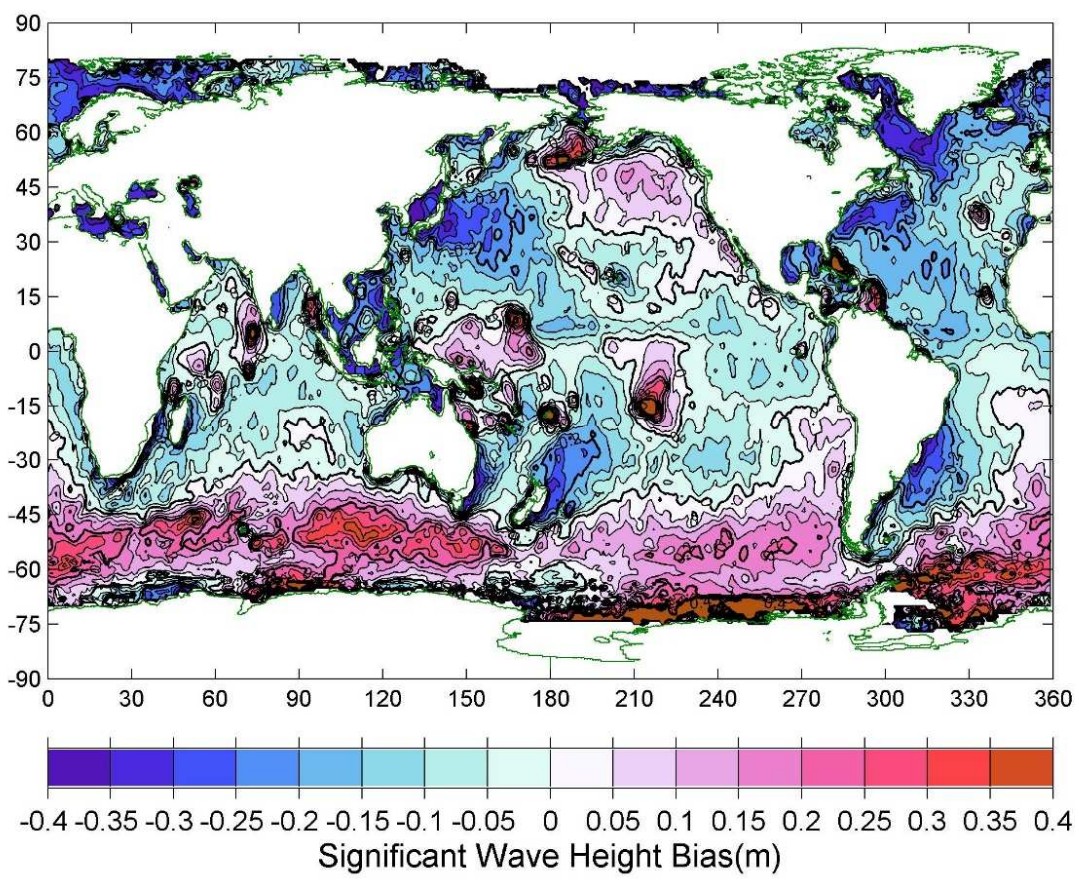

**Figure 6: Bias in significant wave height from the hindcast compared with satellite altimeter measurements, over the period August 1991 – December 2016.**

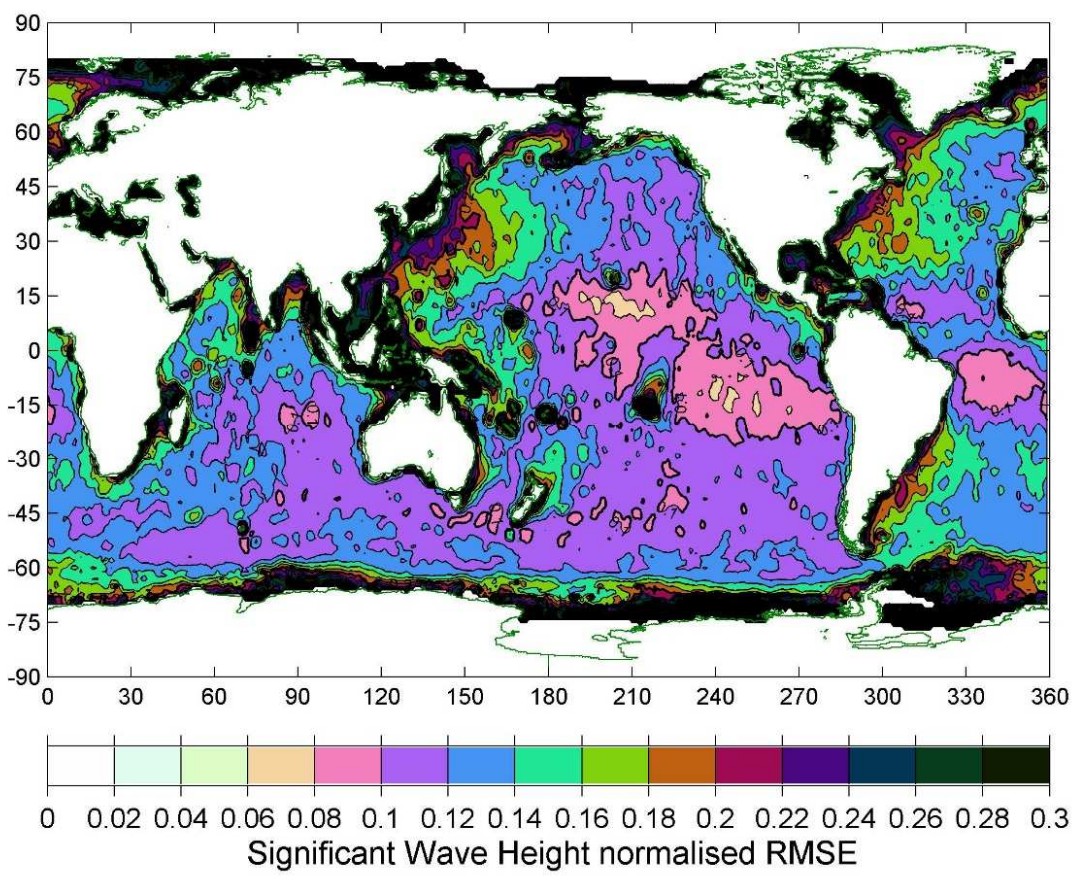

**Figure 7: Normalised root-mean-square error in significant wave height from the hindcast compared with satellite altimeter measurements, over the period August 1991 – December 2016.**

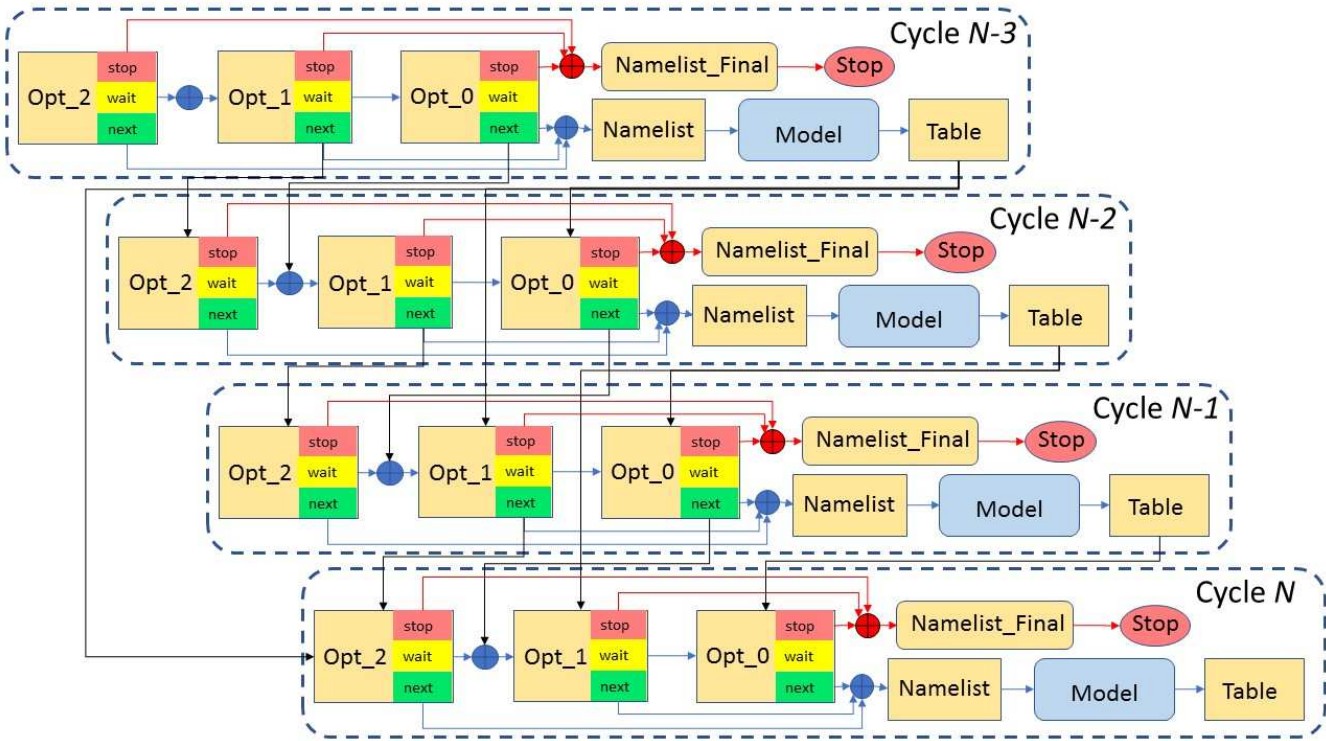

**Figure 8: Dependency graph for the Cyclops optimisation suite, configured to use dependencies to allow for concurrent simulations. This example shows four successive cycles, for the case in which up to three parallel simulations are allowed. Arrows represent dependency, which in some cases are combined by a logical OR (enclosed "+" symbol). All tasks and explicit dependencies (other than suicide triggers) are shown for cycle *N*, but dependencies on cycles before *N-3* are omitted for clarity.**