# Peer review of "Automated model optimisation using the Cylc workflow engine (Cyclops v1.0)"

_Geoscientific Model Development, 2017_

## Referee Comment (RC1) · J.-H. Alves (Referee) · 26 Oct 2017

The manuscript Automated model optimisation using the Cylc workflow engine (Cyclops v1.0) provides a method for objective optimization of source-term coefficients in NWP models, with an application to wind-wave modeling. The method fills in a long-standing gap in wave model development, whereby typically tuning of source-term coefficients has been made in an ad hoc, trial-and-error-based manner, with high time costs and sometimes questionable effectiveness. The proposed method bridges that gap, being potentially a landmark contribution to improving the quality of wave forecasts at operational centers.

The manuscript is well-written and engaging, concise, and clear. I recommend the paper be accepted after minor revisions. A list of suggestions to that end will be provided in a separate cover.

————————————————————

---

## Referee Comment (RC2) · S. F. B. Tett (Referee) · 14 Nov 2017

**Review of Automated model optimisation using the Cylc workflow engine (Cyclops v 1.0).**
**Simon Tett**

I think a potentially interesting paper that should eventually be published. The paper describes a method to use generic optimisation methods to optimise a wave model. In theory the approach could be used for other models though the paper does not really describe the challenges involved in doing this.

I worry the paper is quite close to be a minimum publishable unit and so I am pushing the authors to do it more work. In essence to show their approach does indeed work. To that end I ask that the authors trial two or more additional algorithms. For purely selfish reasons I would be interested in seeing results of the Gauss-Newton approach trailed in Tett et al, 2013 & Tett et al, 2017. However, I understand that the algorithms available to the authors through the NLopt toolkit do not include this. I think the study would also benefit from doing another study in which they started from extreme parameters and see if they end up in the same local optimum or some other one.

The authors do not really deal with the challenge of interfacing the optimisation algorithm to the model. Simply telling us that they generate a simple namelist which gets passed through to the wave model is insufficient detail. I think it would also help the reader if they provided a bit more detail on how the set of previous cases (and cost function values) are passed around. I've done something similar for HadAM3 and much of the effort was in modifying the model namelist variables. HadAM3 has many namelists, each with several variables spread across a few files.

The authors should describe how concurrency happens. I suspect it depends on the optimisation algorithm. If they found a good solution to that that is worth sharing.

One issue that worried us in Tett et al , 2017 was the effect of noise in the optimisation algorithm. If the evaluations needed to fit the $2^{nd}$ order polynomial in BOBYQA are too close to one another then the difference will largely be chaotic noise. How does the authors approach mitigate against that?

Minor comments

P1, L15 – I don't think the URL belongs in the abstract.

L21 – don't think TM belongs in the abstract (and the text uses (R) ).

P2, L10. Note that Roach et al used the system described in Tett et al, 2017.

P2, L12 – I personally don't like 1 sentence paragraphs. Can this sentence be wrapped into the following or preceding paragraph?

P3, L24 read -> reads

P4, l6 A bit more detail on how Cyclops tasks interact would be useful as I don't see a peer reviewed paper describing it. As the optimisation is implemented with special messages being sent some more discussion on messages would be helpful.

P4, l12 interleaveseveral – insert some spaces

P5, line 14 – agree for cases where cost function is some squared difference then –ve values are reasonable. However, I think in the python world returning None to signal need to generate values would be more natural.

P5, L15 – more detail on how the namelist is generated would be helpful. Looking at the code it looks like the text is simply generated. My experience with the Unified Model is that with multiple namelists in multiple files there is a bit of setup to be done to map optimisation variables to namelist variables (in some cases one optimisation variable can modify multiple namelist variables.) Some models may not use namelists so what would be done in this case?

P5, L25 -- Some more detail on how Cycl iterates would be helpful. I think being explicit (and showing how) that Cycl can run several jobs in parallel would be helpful. I think discussing that in the context of the algorithms would also be helpful. I think many algorithms are coded to work serially so won't make use of the ability to run several model simulations in parallel. But clearly authors report doing this so a bit of discussion would help here.

P7, L7 – cite for the model please and don't see the need for the (R)… But I leave it to GMD editors to decide that.

P7, L35 Note this such a cost function (spatial average RMSE ) gives high weight to shortest spatial features which are close to model grid scale and thus very likely strongly affected by model grid and chaotic variability. This is one reason Tett et al, 20113 & 2017 focused on RMS error of large spatial averages. It is a mystery to me why people continue to focus on spatial average RMSE for model evaluation given the smallest scales are dominated by chaotic variability and thus not strongly related to parameter choice or model fidelity.

P8, L2 – can this be typeset larger – probably display would help. Does the dot mean d/dt? If so I think better to spell it out.

P8, line 35 – surely not **zero** impact. Imagine it is very small.

P8 – I found the discussion on the two different packages rather confusing. The authors should rewrite to make this clearer.

P9, L9 – why 0.02 rather than 0.05 or 0.01? Would algorithm terminate if any parameter changed by less than 0.02 or would all need to have changed by less than 0.02?

P9, L11 – why introduce two more parameters?

P9, L24 a bit more discussion about parameter sensitivity here would be useful. For which parameters is the cost function most sensitive?

Table 2 would benefit from some description of the parameters – what do they represent? I don't think readers need to know about "n". It is an implementation detail. Table should also explain what the bold labels are – perhaps better to break up into multiple tables with titles given by meaning of bold labels.

Tables 3&4 – only show parameters that were modified. This would reduce the size considerably and make them less confusing.

Figure 1 – text is small and unreadable (and I don't think the colour is necessary). I suggest just showing one iteration of the work flow with some arrows showing the work flow looping back.

References

Simon F. B. Tett, Michael J. Mineter, Coralia Cartis, Daniel J. Rowlands, and Ping Liu. Can top of atmosphere radiation measurements constrain climate predictions? part 1: Tuning. J. Climate, 26:9348–9366, 2013. doi: 10.1175/JCLID-12-00595.1.

Simon F. B. Tett, Kuniko Yamazaki, Michael J. Mineter, Coralia Cartis, and Nathan Eizenberg. Calibrating climate models using inverse methods: Case studies with HadAM3, HadAM3P and HadCM3. Geoscientific Model Development, 10:3567–3589, September 2017. doi: 10.5194/gmd-2016-305.

---

## Author Comment (AC1) · 20 Nov 2017

The reviewer has made some very constructive comments and suggestions, which we believe can be addressed, leading to an improved manuscript.

It is suggested that the paper needs to be expanded by trialling two or three more additional algorithms, and also by including a study starting from a more extreme parameter set. We accept that suggestion, and intend to undertake some further studies to include in the paper. Our aim was to introduce Cyclops as a potentially useful tool to apply any of a wide set of algorithms to optimise a numerical modelling system, in a way that can be quite readily adapted to different modelling systems. That still

leaves a lot of work to do in designing the modelling system itself (e.g. designing verification metrics, selecting parameters to optimise and choosing the best optimisation algorithm). We did not intend those issues to be a major focus of the paper, and took the view that it would suffice to present one example application for which we made a set of choices that was seen as reasonable if not definitive. But in hindsight, it would help support our aim if we were to show several different optimisation algorithms being applied within Cyclops.

Ideally, if we included at least one truly global algorithm in these further tests, this could go a long way toward clarifying whether we have located a global rather than merely local optimum to our wave hindcast calibration study. In the present version of the paper we merely discussed this issue as possibly needing further investigation. The required computational resources are, however, likely to be prohibitive, so we may need to confine ourselves to local algorithms, and following the suggestion of trying some initialisations far from the default as a simpler way to at least partially address this question. Among local algorithms, we would be amenable to trying the Gauss-Newton approach, but would prefer not to hold up the work to add it to the set of choices available in Cyclops through the NLopt toolbox, if this proves problematic.

The other comments can be addressed through changes in the text. While a revised draft needs to await completion of the further studies noted above, I can make a few points on how we intend to address those comments.

1. The interface between the optimisation algorithm and the model

As noted in the review, the optimisation suite simply outputs a single namelist file. This contains names and values for each parameter, which can be grouped by related sets of parameters. The variable names and allowed ranges are set in a "parameter definition" file that the user prepares. The model suite then needs to include a task that takes this namelist file as one of its inputs, and prepares whatever input files are needed for the model(s) to run. Because the formats are highly model-specific, this task needs to

be tailored for the particular model suite. For example, in our wave hindcast application, this task consists of a shell script which simply includes the namelist file verbatim as part of an ASCII control file, which also has various timing parameters provided from environment variables. In this case, Wavewatch interprets the namelist groups as referring to sets of parameters for different physical processes (e.g. wind input, nonlinear interactions), and we don't need to parse this information in the preprocessing step.

Incidentally, the bold labels in Tables 2-4 refer to these groupings, which was not made sufficiently clear in the paper. Breaking up the tables according to these groupings, as suggested, would also be helpful.

In other cases, some slightly more complicated scripting may be required to generate model inpjut files from the single namelist file. For example, I am presently working on a coupled ADCIRC-SWAN model for hydrodynamic & wave simulations. This model runs as a single coupled executable, but separate SWAN and ADCIRC control files are needed. The cylc suite which controls this coupled simulation includes tasks to prepare those files, consisting of shell scripts using parameter values from environment variables. To apply the optimisation suite to this coupled simulation (which I haven't quite done yet!) will just need to add in a task that parses a single namelist file to set the relevant environmental variables. In that example, there might be "SWAN" and "ADCIRC" namelist groups to make that straightforward. I'm not experienced with HadAM3, but imagine that something similar could be done to prepare the control files it needs. But I guess if the namelist format was inadequate to supply the needed information, we could change that format within the optimisation suite. I should stress that none of this needs any change to the main model codes: they can run as standard release versions under a separate task within the model suite.

When it comes to computing the cost function values, again the details are up to the model suite, but communication with the optimisation suite is very simple: some task within the model suite needs to write that single number to a file, which the optimisation suite reads when that particular implementation of the model suite has completed. The

Interactive
comment

optimisation suite then appends that value, along with the corresponding parameter values, to a simple ASCII file which serves as the "lookup table".

Really, from the optimisation suite's point of view, the model suite is just a black box that takes a namelist file with parameter names and values, written to a specified path relative to a new directory created for each iteration of the model suite, and computes a single objective function value which it writes to another specified path within that directory.

2. Concurrency

The description in Section 2.3 of how Cyclops can run several iterations of the model suite concurrently is perhaps not as clear as it could be. As noted, whether or not concurrent simulations can be run at any particular stage of the optimisation process does indeed depend on the particular algorithm being used. Also, any particular execution of the optimisation algorithm is done in a purely serial way.

Because of the particular generic "user supplied objective function" subroutine we have implemented, any run of the optimisation task ("optimise_step") simply amounts to reading previous results from the lookup table file and deciding, based on those results, what parameter set the particular algorithm would call for next. At that point it stops, and, unless convergence has been reached, those parameter values are sent off to our model "black box", and eventually the lookup table will be appended with the results. So the next time the optimisation task is run, it will get one step further in the iteration sequence.

Now instead of just waiting for the latest model run to finish (in maybe an hour or more), we might wonder if the answer that comes back will make any difference to the next parameter set the optimisation engine will call for. Rather than rely on knowing any details on how each algorithm works, we can decide that empirically, running the optimisation process several times with randomly generated "answers" from the simulation still in progress appended to the lookup table. If it still calls for exactly the same choice

of parameters each time, clearly there is no need to wait for the actual value to be computed, as we already know what simulation needs to be run, we can start it straight away.

Now once we have more than one simulation running, we need to know if the results of any of those runs will affect the selection of the next parameter set. So we keep track of all the parameter values which are currently being worked on, and do randomised tests on sensitivity to those results.

3. Effect of noise in the optimisation algorithm, and choice of cost function

Tett et al (2017) point out that the inherently chaotic nature of the climate system means that a certain level of noise is introduced into evaluations of an atmospheric model simulation, which can cause problems in evaluating the termination criteria. They describe a procedure to rerun a simulation that had nominally satisfied the prescribed convergence criteria, with randomised perturbations before determining whether or not to terminate.

Unlike the atmosphere, ocean surface waves are an essentially dissipative system, and perturbations introduced in the initial conditions and forcing will tend to diminish, rather than grow, with time. As a result, noise in the objective function was not so relevant for our wave hindcast application as for atmospheric models. Nevertheless, we can envision Cyclops being applied to optimisation of an atmospheric model, or some other system with un underlying chaotic nature. So we will add a comment on this issue, suggesting that a measure such as that described by Tett et al (2017) could be introduced into Cyclops for use in such applications. Other references to Tett et al. (2017), which was published subsequent to our manuscript submission, will be added as appropriate e.g. in addition to our reference to Roach et al (2017).

Similarly, the dissipative nature of ocean waves means that a cost function based on a spatial average of the (temporal) RMSE of model-data comparisons will not be subject to the level of chaotic variability seen in similar measures for atmospheric models.

Small scale variability in wave model output is therefore more likely to be genuinely sensitive to parameter variation. In that case it is worth capturing such variability in the cost function, whereas for a chaotic system it may be wiser to average out such variability before evaluating the cost function. Again, we agree it would be helpful to mention this issue in the context that applications to different model systems may require variations in approach.

4. Task interaction

The way that task interaction is handled could be better described. This function is inherent to Cylc, so belongs in its description (Section 2.1). Essentially, Cylc maintains a database for each suite, keeping track of the status of each cycle of each task within the suite. "Status" includes whether the task has started, is running, is stopped, and any messages that the task has sent. Tasks within a Cylc suite can interrogate the suite database, and the databases of other running suites, as configured when task dependencies are defined.

Incidentally, it would perhaps also be useful for us to say a little more about the nomenclature, pointing out the subtle spelling differences. "Cylc" (pronounced "silk") was named partly in a slightly modified reference to the "cycle" concept. "Cyclops" is a Cylc-based Optimisation Suite, but we switched the letters back to sound a little more familiar!

5. Selection of convergence criteria

In the convergence criteria, the "change in parameter values" means the magnitude of the vector difference, i.e. $\sqrt{(\sum_{(n=1)}^{N}(\Delta x\_n)^2}$ ). Section 2.2 will be changed to clarify this. The choice of 0.02 was somewhat arbitrary initial choice. As we noted in Section 3.3, the suite can be restarted with revised criteria after stopping, either having met an initial set of convergence criteria, or through manual intervention. So in practice, you can start with quite loose criteria, then either decide to carry on further with tighter criteria, or reconfigure something and start again. For the two three-month

hindcasts, the aim was to explore differences between the two model configurations, and guide the choice of settings to use for the more thorough 12 month optimisation. For that purpose, the results reached with the 0.02 criterion may have been sufficient, but perhaps it would be better serve the paper to plot the results with an extended iteration under tighter criteria.

6. Tables

The parameters in Table 2 are described in the Appendix, but a reference to this needs to be made in the Table caption. The "n" values were retained in the Tables with the intention of providing a key to the Figures. A clearer alternative way to identify which variable is which in the Figures will be investigated.

Parameters with fixed, default values were added to the Tables for completeness (for other wave modellers who might ask "what value did you use for X?"), but they could indeed be better given elsewhere, if at all.

7. Figures

Figure 1 was generated with software that forms part of the Cylc GUI. It should be straightforward to redo these with improved fonts and colouring. Arrows represent task dependencies, as explicitly specified in defining the suite, so redrawing them to loop back to the start of a cycle would violate this meaning. In Cylc, each task at each cycle can be considered independently, and can run as soon as those explicit dependencies are met. This can allow tasks from one cycle to run simultaneously with tasks from other cycles if their explicit dependencies allow it. This is a feature we exploit in allowing for concurrent model simulations. Actually, a clearer version of Figure 1 could help illustrate that discussion. So my preference is to do that, with other software if necessary, and retain the meaning of the arrows.

8. Minor comments not addressed above

L21, P7 L21: Inconsistencies in referencing the Wavewatch model will be addressed,

in line with the model's licensing terms.

P5, L14. Agreed, it would be more robust and "pythonic" to return "None" than -1 in such cases, so I will change the code before any applications where it could matter.

P8, L2. Yes, dot means d/dt. We will clarify this in the text and improve the equation layout

P8, L35. We will change "zero" to "negligible"

P8 We will rewrite this section in a hopefully clearer manner

P9, L11. The two ice parameters might, a priori, be expected to have more influence than some of the other parameters that were already included, which indeed turned out to be the case. In hindsight they should have been included from the start.

P9, L24. More discussion of parameter sensitivity will be added, using the Delta parameter in Table 4

Other points are minor edits which will be implemented as suggested.

---

## Author Response (AR1)

**Automated model optimisation using the Cylc workflow engine (Cyclops v1.0) – Reviewers' comments**

**J.-H. Alves (Referee 1)**

The manuscript Automated model optimisation using the Cylc workflow engine (Cyclops v1.0) provides a method for objective optimization of source-term coefficients in NWP models, with an application to wind-wave modeling. The method fills in a longstanding gap in wave model development, whereby typically tuning of source-term coefficients has been made in an ad hoc, trial-and-error-based manner, with high time costs and sometimes questionable effectiveness. The proposed method bridges that gap, being potentially a landmark contribution to improving the quality of wave forecasts at operational centers.

The manuscript is well-written and engaging, concise, and clear. I recommend the paper be accepted after minor revisions. A list of suggestions to that end will be provided in a separate cover.

**Simon Tett (Referee 2)**

I think a potentially interesting paper that should eventually be published. The paper describes a method to use generic optimisation methods to optimise a wave model. In theory the approach could be used for other models though the paper does not really describe the challenges involved in doing this.

I worry the paper is quite close to be a minimum publishable unit and so I am pushing the authors to do it more work. In essence to show their approach does indeed work. To that end I ask that the authors trial two or more additional algorithms. For purely selfish reasons I would be interested in seeing results of the Gauss-Newton approach trailed in Tett et al, 2013 & Tett et al, 2017. However, I understand that the algorithms available to the authors through the NLopt toolkit do not include this. I think the study would also benefit from doing another study in which they started from extreme parameters and see if they end up in the same local optimum or some other one.

The authors do not really deal with the challenge of interfacing the optimisation algorithm to the model. Simply telling us that they generate a simple namelist which gets passed through to the wave model is insufficient detail. I think it would also help the reader if they provided a bit more detail on how the set of previous cases (and cost function values) are passed around. I've done something similar for HadAM3 and much of the effort was in modifying the model namelist variables. HadAM3 has many namelists, each with several variables spread across a few files.

The authors should describe how concurrency happens. I suspect it depends on the optimisation algorithm. If they found a good solution to that that is worth sharing.

One issue that worried us in Tett et al , 2017 was the effect of noise in the optimisation algorithm. If the evaluations needed to fit the $2_{nd}$ order polynomial in BOBYQA are too close to one another then the difference will largely be chaotic noise. How does the authors approach mitigate against that?

Minor comments

P1, L15 – I don't think the URL belongs in the abstract.

L21 – don't think TM belongs in the abstract (and the text uses (R) ).

P2, L10. Note that Roach et al used the system described in Tett et al, 2017.

P2, L12 – I personally don't like 1 sentence paragraphs. Can this sentence be wrapped into the following or preceding paragraph?

P3, L24 read -> reads

P4, l6 A bit more detail on how Cyclops tasks interact would be useful as I don't see a peer reviewed paper describing it. As the optimisation is implemented with special messages being sent some more discussion on messages would be helpful.

P4, l12 interleaveseveral – insert some spaces

P5, line 14 – agree for cases where cost function is some squared difference then –ve values are reasonable. However, I think in the python world returning None to signal need to generate values would be more natural.

P5, L15 – more detail on how the namelist is generated would be helpful. Looking at the code it looks like the text is simply generated. My experience with the Unified Model is that with multiple namelists in multiple files there is a bit of setup to be done to map optimisation variables to namelist variables (in some cases one optimisation variable can modify multiple namelist variables.) Some models may not use namelists so what would be done in this case?

P5, L25 -- Some more detail on how Cycl iterates would be helpful. I think being explicit (and showing how) that Cycl can run several jobs in parallel would be helpful. I think discussing that in the context of the algorithms would also be helpful. I think many algorithms are coded to work serially so won't make use of the ability to run several model simulations in parallel. But clearly authors report doing this so a bit of discussion would help here.

P7, L7 – cite for the model please and don't see the need for the (R)... But I leave it to GMD editors to decide that.

P7, L35 Note this such a cost function (spatial average RMSE ) gives high weight to shortest spatial features which are close to model grid scale and thus very likely strongly affected by model grid and chaotic variability. This is one reason Tett et al, 20113 & 2017 focused on RMS error of large spatial averages. It is a mystery to me why people continue to focus on spatial average RMSE for model evaluation given the smallest scales are dominated by chaotic variability and thus not strongly related to parameter choice or model fidelity.

P8, L2 – can this be typeset larger – probably display would help. Does the dot mean d/dt? If so I think better to spell it out.

P8, line 35 – surely not **zero** impact. Imagine it is very small.

P8 – I found the discussion on the two different packages rather confusing. The authors should rewrite to make this clearer.

P9, L9 – why 0.02 rather than 0.05 or 0.01? Would algorithm terminate if any parameter changed by less than 0.02 or would all need to have changed by less than 0.02?

P9, L11 – why introduce two more parameters?

P9, L24 a bit more discussion about parameter sensitivity here would be useful. For which parameters is the cost function most sensitive?

Table 2 would benefit from some description of the parameters – what do they represent? I don't think readers need to know about "n". It is an implementation detail. Table should also explain what the bold labels are – perhaps better to break up into multiple tables with titles given by meaning of bold labels.

Tables 3&4 – only show parameters that were modified. This would reduce the size considerably and make them less confusing.

Figure 1 – text is small and unreadable (and I don't think the colour is necessary). I suggest just showing one iteration of the work flow with some arrows showing the work flow looping back.

References

Simon F. B. Tett, Michael J. Mineter, Coralia Cartis, Daniel J. Rowlands, and Ping Liu. Can top of atmosphere radiation measurements constrain climate predictions? part 1: Tuning. J. Climate, 26:9348–9366, 2013. doi: 10.1175/JCLID-12-00595.1.

Simon F. B. Tett, Kuniko Yamazaki, Michael J. Mineter, Coralia Cartis, and Nathan Eizenberg. Calibrating climate models using inverse methods: Case studies with HadAM3, HadAM3P and HadCM3. Geoscientific Model Development, 10:3567–3589, September 2017. doi: 10.5194/gmd-2016-305.

**Automated model optimisation using the Cylc workflow engine (Cyclops v1.0)**
**Response by to review comments**

The comments by Reviewer #1 are covered by our response to those of Reviewer #2, which we detail below.

The reviewer has made some very constructive comments and suggestions, which we believe can be addressed, leading to an improved manuscript.

It is suggested that the paper needs to be expanded by trialling two or three more additional algorithms, and also by including a study starting from a more extreme parameter set. We accept that suggestion, and have undertakes some further studies to include in the paper. Our aim was to introduce Cyclops as a potentially useful tool to apply any of a wide set of algorithms to optimise a numerical modelling system, in a way that can be quite readily adapted to different modelling systems. That still leaves a lot of work to do in designing the modelling system itself (e.g. designing verification metrics, selecting parameters to optimise and choosing the best optimisation algorithm). We did not intend those issues to be a major focus of the paper, and took the view that it would suffice to present one example application for which we made a set of choices that was seen as reasonable if not definitive. But in hindsight, it would help support our aim if we were to show several different optimisation algorithms being applied within Cyclops.

We have included one global algorithm in these further tests, along with tests with a local algorithm starting from a series of different, randomly chosen, initial parameter sets. We believe this has helped clarifying whether we have located a global rather than merely local optimum to our wave hindcast calibration study. We did not, however, find it feasible to spend the necessary time implementing the Gauss-Newton approach into the NLopt toolbox.

The other comments were addressed through changes in the text. On the principal points:

1. **The interface between the optimisation algorithm and the model**

As noted in the review, the optimisation suite simply outputs a single namelist file. This contains names and values for each parameter, which can be grouped by related sets of parameters. The variable names and allowed ranges are set in a "parameter definition" file that the user prepares. The model suite then needs to include a task that takes this namelist file as one of its inputs, and prepares whatever input files are needed for the model(s) to run. Because the formats are highly model-specific, this task needs to be tailored for the particular model suite. For example, in our wave hindcast application, this task consists of a shell script which simply includes the namelist file verbatim as part of an ASCII control file, which also has various timing parameters provided from environment variables. In this case, Wavewatch interprets the namelist groups as referring to sets of parameters for different physical processes (e.g. wind input, nonlinear interactions), and we don't need to parse this information in the preprocessing step.

In other cases, some slightly more complicated scripting may be required to generate model input files from the single namelist file. We have covered this requirement more extensively in the revised manuscript, as well as stressing that no change is needed to the main model codes

When it comes to computing the cost function values, again the details are up to the model suite, but communication with the optimisation suite is very simple: some task within the model suite needs to write that single number to a file, which the optimisation suite reads when that particular implementation of the model suite has completed. The optimisation suite then appends that value, along with the corresponding parameter values, to a simple ASCII file which serves as the "lookup table". Really, from the optimisation suite's point of view, the model suite is just a black box that takes a namelist file with parameter names and values, written to a specified path relative to a new directory created for each iteration of the model suite, and computes a single objective function value which it writes to another specified path within that directory.

2. **Concurrency**

There are perhaps three points which need to be made in describing how Cyclops allows for concurrent simulations.

1. How the optimisation algorithm can decide whether a new parameter set could be evaluated while waiting for other function evaluations (i.e. model runs) to complete.

2. How Cylc in general allows for tasks from different cycles to execute concurrently
3. The specific way that Cyclops uses that ability to allow for concurrent model evaluations

The way we have

The description in Section 2.3 of how Cyclops can run several iterations of the model suite concurrently has been considerably rewritten. As noted, whether or not concurrent simulations can be run at any particular stage of the optimisation process does indeed depend on the particular algorithm being used. Also, any particular execution of the optimisation algorithm is done in a purely serial way.

On the first point, we have tried to give a clearer description of how the "optimise" task works. Because of the particular generic "user supplied objective function" subroutine we have implemented, any run of the optimisation task simply amounts to reading previous results from the lookup table file and deciding, based on those results, what parameter set the particular algorithm would call for next. If there are "active" parameter sets still being evaluated, it can do a set of tests with random "answers" to those active evaluations, to decide if that affects the value of the parameters it calls for.

The second point has been addressed by improving the description of Cylc in section 2.1, to emphasise that tasks will execute whenever their dependencies allow, and do not necessarily iterate in a rigid cycle.

On the third point, we have attempted to provide a clearer description, aided by replacing the old Figure 1 with some clearer diagrams of task dependencies. In fact, we realised that there is a simpler way to implement concurrency in Cyclops, letting tasks retry after failure, rather than using the somewhat complex dependency structure we originally implemented. Having determined that the two methods achieve the same result, we describe the new method in the body of the text, so that readers can more readily grasp the important concepts. The original method is relegated to an Appendix.

**3. Effect of noise in the optimisation algorithm, and choice of cost function**

Tett et al (2017) point out that the inherently chaotic nature of the climate system means that a certain level of noise is introduced into evaluations of an atmospheric model simulation, which can cause problems in evaluating the termination criteria. They describe a procedure to rerun a simulation that had nominally satisfied the prescribed convergence criteria, with randomised perturbations before determining whether or not to terminate.

Unlike the atmosphere, ocean surface waves are an essentially dissipative system, and perturbations introduced in the initial conditions and forcing will tend to diminish, rather than grow, with time. As a result, noise in the objective function was not so relevant for our wave hindcast application as for atmospheric models. Nevertheless, we can envision Cyclops being applied to optimisation of an atmospheric model, or some other system with un underlying chaotic nature. So have added a comment to that effect, suggesting that a measure such as that described by Tett et al (2017) could be introduced into Cyclops for use in such applications. Other references to Tett et al. (2017), which was published subsequent to our manuscript submission, have been added.

Similarly, the dissipative nature of ocean waves means that a cost function based on a spatial average of the (temporal) RMSE of model-data comparisons will not be subject to the level of chaotic variability seen in similar measures for atmospheric models. Small scale variability in wave model output is therefore more likely to be genuinely sensitive to parameter variation. In that case it is worth capturing such variability in the cost function, whereas for a chaotic system it may be wiser to average out such variability before evaluating the cost function. We have added a mention this issue in the context that applications to different model systems may require variations in approach.

**4. Task interaction**

The way that task interaction is handled could be better described. This function is inherent to Cylc, so its description in Section 2.1 has been improved.

**5. Selection of convergence criteria**

In the convergence criteria, the "change in parameter values" means the magnitude of the vector difference, i.e. $\sqrt{\sum_{n=1}^{Npar}(\Delta x_n)^2}$. Section 2.2 was be changed to clarify this. The choice of 0.02 (in the original manuscript) was somewhat arbitrary initial choice. As we noted in Section 3.3, the suite can be restarted with revised criteria after stopping, either having met an initial set of convergence criteria, or through manual intervention. So in practice, you can start with quite loose criteria, then either decide to carry on further with tighter criteria, or reconfigure something and start again. For the two three-month hindcasts, the aim was to explore differences between the two model configurations, and guide the choice of settings to use for the more thorough 12 month optimisation. For that purpose, the results reached with the 0.02 criterion may have been sufficient, but we have now extended the iterations until stricter criteria (0.0001 fractional change) have been met.

**6. Tables**

The parameters in Table 2 are now more clearly referenced in the Table caption, which also mentions that the "n" values provide a key to the Figures.

Parameters with fixed, default values were originally added to the Tables for completeness, but they have now been removed.

**7. Figures**

Figure 1 has been replaced by two clearer Figures

**8. Minor comments not addressed above**

L21, P7 L21: Inconsistencies in referencing the Wavewatch model have been addressed, in line with the model's licensing terms.

P5, L14. Agreed, it would be more robust and "pythonic" to return "None" than -1 in such cases. This does not cause a problem in the present application, but the point has been mentioned, and will be implemented in the code in future.

P8, L2. Yes, dot means d/dt. This has been clarify in the text, and the equation layout improved

P8, L35. We have changed "zero" to "negligible"

P8 This section has been rewritten in a hopefully clearer manner

P9, L11. The two ice parameters might, a priori, be expected to have more influence than some of the other parameters that were already included, which indeed turned out to be the case. In hindsight they should have been included from the start.

P9, L24. More discussion of parameter sensitivity has been added, using the Delta parameter in Table 4

Other points are minor edits which have been implemented as suggested.

[revised manuscript text omitted]

---

## Referee Report (RR1)

Review of  Automated model optimisation using the Cylc workflow engine (Cyclops v1.0)
By Gorman and Oliver

Revisions are good and the paper can be published subject to some more minor revisions. I think the explanation of section 2.3 could be improved to clarify how the concurrent executions are done. The idea is neat so is worth explaining well so others can see it too.

I put the following figure together to help me understand the approach. Suspect it is too much for the paper but something like it would help some readers assuming it is correct.

Aim: Identify parameter values that only depend on current state and previous iterations.

Random number seed

| p(1.1) with random f(p(1.1)) | p(2.1) with random f(p(2.1)) | p(3.1) with random f(p(3.1)) | p(4.1) with random f(p(4.1)) | Run if ALL p(X.1) the same |
| p(1.2) with random f(p(1.2)) | p(2.2) with random f(p(2.2)) | p(3.2) with random f(p(3.2)) | p(4.2) with random f(p(4.2)) | Run if ALL p(X.2) the same |
| p(1.3) with random f(p(1.3)) | p(2.3) with random f(p(2.3)) | p(3.3) with random f(p(3.3)) | p(4.3) with random f(p(4.3)) | Run if ALL p(X.3) the same |
| p(1.4) with random f(p(1.4)) | p(2.4) with random f(p(2.4)) | p(3.4) with random f(p(3.4)) | p(4.4) with random f(p(4.4)) | Run if ALL p(X.4) the same |
| p(1.5) with random f(p(1.5)) | p(2.5) with random f(p(2.5)) | p(3.5) with random f(p(3.5)) | p(4.5) with random f(p(4.5)) | Run if ALL p(X.5) the same |

Function evaluations

p(X.Y) is a function of all earlier parameter values

---

## Author Response (AR2)

**Automated model optimisation using the Cylc workflow engine (Cyclops v1.0) – Response to Reviewers' comments (round 2)**

The reviewer commented:

5 "Revisions are good and the paper can be published subject to some more minor revisions. I think the explanation of section 2.3 could be improved to clarify how the concurrent executions are done. The idea is neat so is worth explaining well so others can see it too.
I put the following figure together to help me understand the approach. Suspect it is too much for the paper but something like it would help some readers assuming it is correct."

Our response:

We thank the reviewer for the attention he has given our manuscript.

15 In response, we have revised the text, principally section 2.3 on concurrency, apart from adding an equation (2) earlier on to help get across the concept of what the "Optimise" task actually does.

The revised text addresses any confusion about the tests run with randomised values for objective function (f) values that are still being evaluated, being more explicit about how these are run. We have tried to emphasise
20 that these multiple randomised tests are a serial process that happens within each Optimise task, to determine if there is a single definitive parameter set for that cycle to proceed with. We also try to make it clearer that those multiple tests are not being run in parallel.

The reviewer's suggested figure did not quite represent the random testing process that we actually use, but
25 with the expanded explanation didn't consider that any revised version of that suggested Figure would add further clarification. However we did add a new Figure (3) after the description of the Cylc implementation of concurrency (retry version) to talk through an example of how that might work in a test case.

[revised manuscript text omitted]